# Neurexin-3 subsynaptic densities are spatially distinct from Neurexin-1 and essential for excitatory synapse nanoscale organization in the hippocampus

Brian A Lloyd [1], Ying Han[2,3], Rebecca Roth [1], Bo Zhang [2,3] & Jason Aoto [1] ✉

Proteins critical for synaptic transmission are non-uniformly distributed and assembled into regions of high density called subsynaptic densities (SSDs) that transsynaptically align in nanocolumns. Neurexin-1 and neurexin-3 are essential presynaptic adhesion molecules that non-redundantly control NMDAR- and AMPAR-mediated synaptic transmission, respectively, via transsynaptic interactions with distinct postsynaptic ligands. Despite their functional relevance, fundamental questions regarding the nanoscale properties of individual neurexins, their influence on the subsynaptic organization of excitatory synapses and the mechanisms controlling how individual neurexins engage in precise transsynaptic interactions are unknown. Using Double Helix 3D dSTORM and neurexin mouse models, we identify neurexin-3 as a critical presynaptic adhesion molecule that regulates excitatory synapse nano-organization in hippocampus. Furthermore, endogenous neurexin-1 and neurexin-3 form discrete and non-overlapping SSDs that are enriched opposite their postsynaptic ligands. Thus, the nanoscale organization of neurexin-1 and neurexin-3 may explain how individual neurexins signal in parallel to govern different synaptic properties.

To effectively process information, presynaptic release machinery and postsynaptic receptors are tightly regulated. Super-resolution microscopy has revealed that proteins critical for presynaptic release and postsynaptic detection of neurotransmitters are distributed in a non-uniform manner and assemble into regions of high density, called subsynaptic densities (SSDs), which are transsynaptically aligned to form nanocolumns. The excitatory synaptic nanocolumn is commonly defined by the transsynaptic alignment of presynaptic Rab3 interacting molecules (RIMs) SSDs opposite postsynaptic AMPAR and/or PSD-95 SSDs, however, an understanding of the mechanisms underlying SSD formation and transsynaptic nanocolumnar alignment is incomplete[1–5]. Cell-adhesion molecules may control synaptic nanoscale architecture because they physically connect pre- and post-synapses and recruit protein complexes to sites of synaptic transmission. Presynaptic neurexins (Nrxns) are potential candidates because they control critical aspects of synaptic transmission.

Nrxns are evolutionarily conserved presynaptic adhesion molecules that are essential for synapse function[6]. Three *Nrxn* genes (*Nrxn1-3*) encode for longer α and shorter β forms and a Nrxn1-specific γ product[7]. Additionally, Nrxns are subject to extensive alternative splicing at six conserved splice sites (SS1-6). Although the biological significance of Nrxn alternative splicing is poorly understood, the inclusion (SS4+) or exclusion (SS4−) of the alternative exon regulates binding to most postsynaptic ligands to control synapse function[7].

[1]University of Colorado Anschutz School of Medicine, Department of Pharmacology, Aurora, CO 80045, USA. [2]School of Chemical Biology and Biotechnology, Peking University Shenzhen Graduate School, Shenzhen 518055, China. [3]Institute of Neurological and Psychiatric Disorders, Shenzhen Bay Laboratory, Shenzhen 518132, China. ✉e-mail: jason.aoto@cuanschutz.edu

Individual Nrxns dominantly govern different aspects of synapse function in a brain-region- and synapse-specific manner at excitatory[8–12] and inhibitory synapses[13–15]. At the same excitatory hippocampal synapses, Nrxn1 controls NMDA receptor currents while *Nrxn3* controls AMPA receptor strength[8,10]. Recent work has revealed that Nrxn1 SS4+ and Nrxn3 SS4+ can form tripartite complexes with cerebellin-2 (Cbln2) and GluD1. Functioning through different signaling pathways, Nrxn1-Cbln2-GluD1 stabilizes synaptic NMDA receptors, while Nrxn3-Cbln2-GluD1 destabilizes AMPA receptors[12]. Furthermore, the deletion of Nrxn3 from hippocampal neurons decreases AMPAR-mediated synaptic transmission by ~40–50%[9]. To counterbalance the destabilizing effects of Nrxn3-Cbln2-GluD1 signaling, Nrxn3 SS4− (the dominant SS4 isoform in hippocampus) likely interacts with postsynaptic leucine-rich repeat protein-2 (LRRTM2) to promote AMPAR surface stability[8]. How do Nrxn1 and Nrxn3 mediate precise transsynaptic signaling and, in the case of Nrxn3, maintain synaptic AMPAR homeostasis? An appealing possibility is that Nrxns and their ligands have a distinct nanoscale architecture to promote and ensure appropriate synapse function.

The ablation of *Nrxn3α* and *Nrxn3β* (*Nrxn3* KO) selectively impairs AMPAR-mediated synaptic transmission by ~40–50% in hippocampal neurons, suggesting that Nrxn3 may be a critical, yet untested, regulator of synaptic nanoscale architecture[9]. The manipulation of Nrxn ligands, including LRRTM2, impairs AMPAR-mediated synaptic transmission and significantly alters AMPAR SSD properties[5,16,17]. Here, we used Double Helix 3D direct stochastic optical reconstruction microscopy (3D dSTORM) and identify presynaptic Nrxn3 as a critical regulator of synaptic nano-organization of excitatory synapses in hippocampus. To directly examine the nanoscale properties of Nrxn1 and Nrxn3, we combined 3D dSTORM with a previously generated *HA-Nrxn1* mouse and developed herein an epitope-tagged *V5-Nrxn3* mouse, which permits the detection of all full-length Nrxn3 isoforms. Nrxn1 SSDs are not highly enriched in excitatory nanocolumns but are peripherally enriched near GluD1 SSDs. By contrast, Nrxn3 assembles into SSDs that are enriched in excitatory nanocolumns and exhibit a nanoscale distribution opposite LRRTM2 SSDs. Lastly, in the same synapse, endogenous Nrxn1 and Nrxn3 SSDs are discrete and non-overlapping. Our results identify an unexpected potential nanoscale mechanism that may contribute to the formation of the "molecular code" that specifies synapse identity and function.

## Results

### Deletion of *Neurexin-3* alters GluA1 nanoscale organization in hippocampal neurons

The genetic ablation of *Nrxn3* selectively and robustly impairs AMPAR-mediated synaptic transmission by ~40–50% without altering presynaptic function[9]. To test whether alterations in the nanoscale organization of excitatory synapses underscores the *Nrxn3* KO AMPAR phenotype, we used Double Helix 3D dSTORM[18] and previously published code[19] to analyze *Nrxn3α/β* cKO neurons (Fig. S1a). Unless noted, all antibodies used in this study were directly conjugated, as the resolution of Double Helix 3D-STORM approaches the maximum distance that could separate an epitope and fluorophore using traditional indirect immunofluorescence[20].

We infected *Nrxn3α/β* cKO primary hippocampal cultures with active (*Nrxn3* KO) or inactive (control) cre-recombinase expressing lentiviruses on day in vitro 4 (DIV4). Lentivirus routinely transduces >95% of all neurons, which is essential to study the role of presynaptic molecules[8]. *Nrxn3* KO neurons have undetectable levels of *Nrxn3* mRNA[9] (Fig. S1b). We performed live surface labeling of the AMPAR GluA1 subunit followed by cell-permeabilized co-staining for the integral active zone protein RIM1. Consistent with previous reports, we observed that the single molecule localizations representing RIM1 and GluA1 were assembled into disc-shaped clusters, which are referred to as synaptic compartments (Fig. 1a). The volume of a synaptic compartment is defined as the minimal volume required to encompass all single-molecule localizations of a given protein[21]. Within synaptic compartments, the distribution of RIM1 and GluA1 were non-uniform and contained multiple regions of high-density regions called sub-synaptic densities (SSDs) (Fig. 1a)[3,22,23]. Using established computational methods to quantitatively assess these nanoscale properties[19], we found that the knockout of *Nrxn3* decreased the GluA1 synaptic compartment volume by 32.2% and the number of GluA1 SSDs by 17.4% but did not alter the volume of GluA1 SSDs (Fig. 1a–d). Further, the deletion of *Nrxn3* decreased the relative GluA1 density inside SSDs by 19.2% and increased the synaptic compartment density of GluA1 by 29.0% (Figs. 1e and S1c). The decreased GluA1 compartment volume is consistent with previous confocal imaging determined that *Nrxn3* KO reduced the size of surface GluA1[9]. Although the density of GluA1 in the synaptic compartment increased, which may appear contrary to the functional *Nrxn3* KO phenotype, it is likely a result of the condensed compartment volume rather than an increase in the absolute amount of GluA1 (as discussed below). Importantly, given the proposed functional importance of SSDs, *Nrxn3* KO reduced the average number of GluA1 SSDs per synapse and decreased GluA1 receptor density within SSDs, which provides insight into a potential mechanism that underlies the Nrxn3-dependent deficit in AMPAR-mediated synaptic transmission.

Unexpectedly, although *Nrxn3* KO in hippocampal primary neurons or ex vivo slice preparations did not alter excitatory presynaptic properties[9], the deletion of *Nrxn3* reduced the compartment volume of RIM1 by 44.2% and the volume of RIM1 SSDs by 38.7%. However, the average number of RIM1 SSDs per synapse and the density of RIM1 molecules inside SSDs were unchanged, indicating that the number of functional release sites and the density of RIM1 at these sites may be sufficient to sustain presynaptic release (Fig. 1f–i). Further, large changes in presynaptic volume alone do not necessarily result in equivalent changes to presynaptic release probability[24]. Perhaps it is not surprising that the ablation of *Nrxn3* altered the nanoscale properties of RIM1 independent of a functional presynaptic phenotype because α-Nrxns redundantly couple calcium channels to control presynaptic release[6]. Together with our GluA1 data (Fig. 1b–e), the change in the pre- and post-synaptic nanoscale organization suggests that Nrxn3 may signal bidirectionally – presynaptically via its intracellular sequences to aid in RIM1 localization and transsynaptically via its extracellular sequences to govern postsynaptic GluA1.

We next utilized a validated protein enrichment analysis to assess whether Nrxn3 regulates the transsynaptic alignment of GluA1 and RIM1[3,19]. Nanocolumn alignment of two proteins would predict that the centroid of an SSD on one side of the synapse should oppose a region of higher normalized protein density on the other. We first quantitatively determined the number of GluA1 molecules within binned distances from the centroid of RIM1. We also randomized the GluA1 localizations and determined the number of randomized GluA1 molecules opposite RIM1 at the same binned distances. Dividing the experimentally determined density distribution by the randomized dataset resulted in the normalized GluA1 density from the centroid of a RIM1 SSD - thus, an enrichment value of 1 represents a randomized distribution of GluA1 molecules (Fig. 1j)[19]. Further, we averaged the normalized GluA1 density within a radius of 60 nm from the centroid of RIM1 to determine the enrichment index (Fig. 1j)[19]. *Nrxn3* ablation significantly reduced the density of GluA1 molecules opposite RIM1 SSDs and reduced the enrichment index by more than 33% (Fig. 1j). By contrast, RIM1 density opposite GluA1 SSDs was unchanged, suggesting a postsynaptic nanoscopic remodeling of GluA1 and that the remaining GluA1 SSDs were still aligned with regions of presynaptic RIM1 enrichment (Fig. S1d, e)[5]. As a complementary approach, we measured the paired cross-correlation between RIM1 and GluA1 density distributions[19]. This analysis is independent of SSD detection and instead identifies the spatial relationship between the density

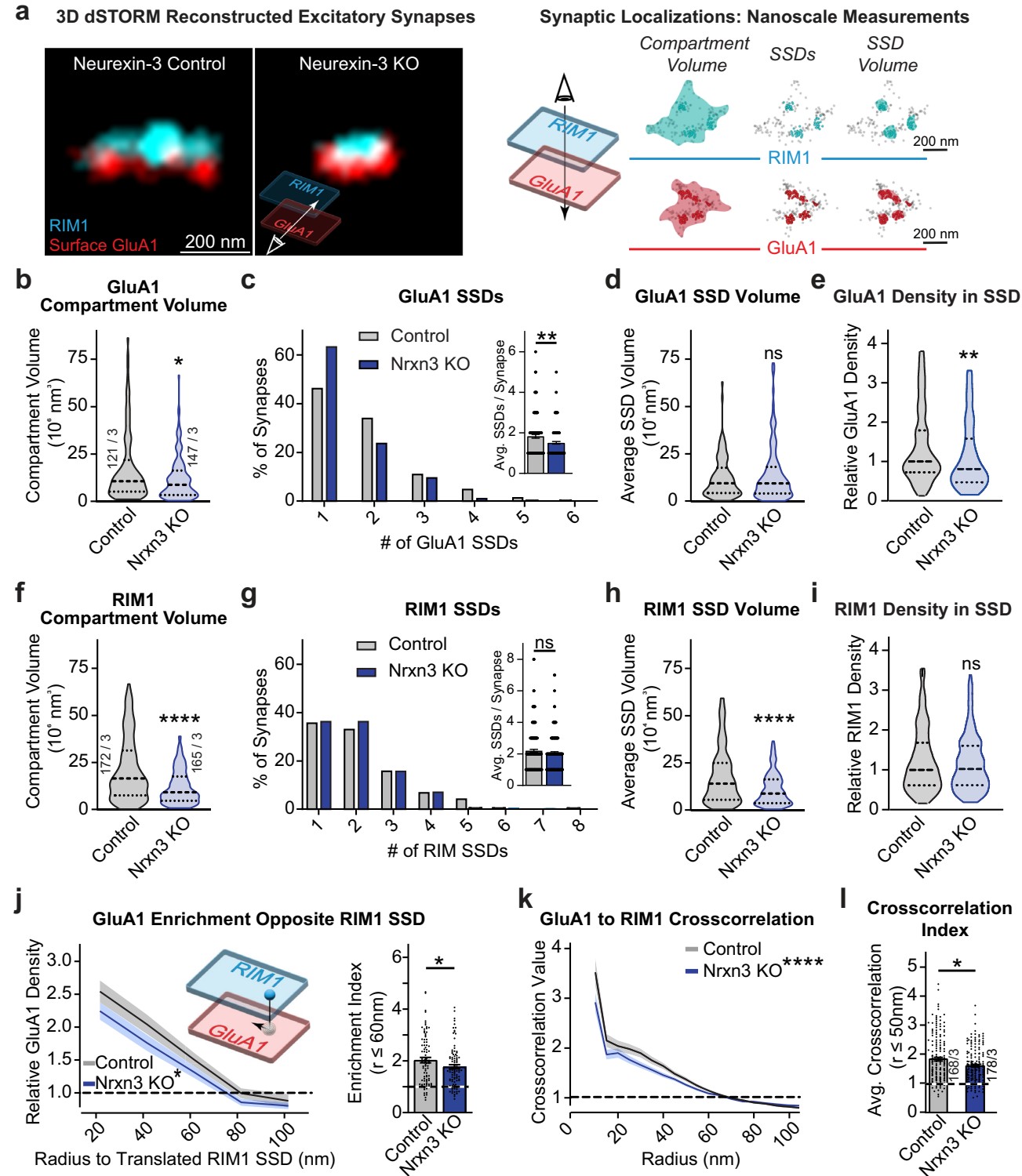

**a** 3D dSTORM Reconstructed Excitatory Synapses

Synaptic Localizations: Nanoscale Measurements

distributions of GluA1 and RIM1. Paired cross-correlation analysis revealed a 28.5% decrease in GluA1 and RIM1 alignment from radii of 10–50 nm, which further supports the notion that the ablation of *Nrxn3* decreases the amount of GluA1 opposite regions of high RIM1 density (Fig. 1k–i). Overall, the enrichment and paired cross-correlation analyses suggest that the decreased density of AMPARs in SSDs after *Nrxn3* KO is functionally relevant as the density of GluA1 single molecule localizations in synaptic nanocolumns is significantly reduced. By contrast, despite a significant reduction in RIM1 compartment volume and SSD volume, the enrichment analysis indicates that the density of

RIM1 opposite GluA1 SSDs was not affected by the deletion of *Nrxn3*, which further supports the notion that presynaptic function is unaltered following the loss of Nrxn3 in hippocampus. Taken together, these data indicate that Nrxn3 plays a critical role in regulating the nanoscale architecture of GluA1 in hippocampus, which is largely consistent with the *Nrxn3* KO AMPAR phenotype and provide a possible nanoscale mechanism to account for the selective deficit of AMPAR-mediated synaptic transmission[9].

Similar to Nrxn3, Nrxn1 is highly expressed in hippocampus, however, unlike Nrxn3, the conditional deletion of *Nrxn1* does not alter

**Fig. 1 | Conditional deletion of *Nrxn3* impairs nano-organization of GluA1 and RIM1 at excitatory synapses in the hippocampus. a** Representative 3D dSTORM images of presynaptic active zone maker RIM1 (cyan) and postsynaptic AMPA receptor subunit GluA1 (red) in Control and *Nrxn3* knockout neurons (left). The schema used to quantify 3D dSTORM data (right). *Nrxn3* KO reduces GluA1 compartment volume $p = 0.0148$ (**b**) and number of SSDs per synapse $p = 0.0045$ $n = 122$ control and 149 *Nrxn3* KO synapses (**c**) but not SSD volume $p = 0.8995$ $n = 115$ and $n = 135$ synapses (**d**). *Nrxn3* KO reduces the relative density of GluA1 in SSDs $p = 0.0098$ n = 103 synapses and $n = 123$ synapses (**e**). Significance: Mann–Whitney test (two-tailed). **f–i** Same quantification as (**b–e**) except for presynaptic RIM1. RIM1 compartment volume, $p < 0.0001$; SSDs per synapse, $p = 0.5016$ $n = 191$ control and $n = 185$ *Nrxn3* KO synapses; SSD volume $p < 0.0001$ $n = 163$ control and $n = 150$ *Nrxn3* KO synapses; relative density in SSDs $p = 0.7732$ $n = 169$ control and $n = 160$

*Nrxn3* KO synapses. Significance: Mann–Whitney test (two-tailed). **j** *Nrxn3* KO reduces transsynaptic enrichment of GluA1 relative to RIM1 SSDs, $p = 0.0150$ $n = 87$ control and $n = 97$ *Nrxn3* KO SSDs and a reduction in the GluA1 transsynaptic enrichment index at radii ≤60 nm, $p = 0.0471$ $n = 87$ control and $n = 97$ *Nrxn3* KO SSDs. Significance: 2-way repeated measures ANOVA main effect of *Nrxn3* KO and Student's $t$ test (two-tailed) respectively. **k, l** *Nrxn3* KO reduces GluA1 to RIM1 cross-correlation over 10–100 nm radius, $p = 0.0001$ $n = 174$ control and $n = 188$ *Nrxn3* KO synapses; (**k**) and average cross-correlation at radii ≤50 nm, $p = 0.0307$ (**l**). Significance: 2-way repeated measures ANOVA main effect of radius x *Nrxn3* KO; Mann–Whitney test (two-tailed). Data from three independent experiments. Number of synapses indicated on the graph unless stated in the legend. Bar graphs and line graphs: average ± SEM. Violin plots: median ± upper and lower quartiles. *$p < 0.05$; **$p < 0.01$. Source data are provided as a Source Data file.

excitatory synapse morphology or AMPAR-mediated synaptic transmission[25]. To test the impact of Nrxn1 on the nanoscale organization of excitatory synapses, we cultured neurons from a validated conditional *HA-Nrxn1* knock-in mouse[25]. In the absence of cre-recombinase, full-length HA-tagged Nrxn1 is produced, however, in the presence of Cre-recombinase, the conditional gene produces a non-functional truncated Nrxn1 protein (Fig. S1f)[25]. Thus, akin to *Nrxn1* KO, Cre-dependent truncation of HA-Nrxn1 eliminates Nrxn1-dependent transsynaptic and intracellular signaling. In contrast to the *Nrxn3* KO phenotype, the inactivation of Nrxn1 did not alter miniature excitatory postsynaptic currents or the nanoscale properties of RIM1 and GluA1 (Fig. S1g–t). Thus, our data identify Nrxn3 as a key presynaptic adhesion molecule required for the nanoscale organization and transsynaptic alignment of GluA1–RIM1 nanocolumns in hippocampal neurons.

### Deletion of *Neurexin-3* alters PSD-95 nanoscale organization in hippocampal neurons

The functional and nanoscale AMPAR phenotype observed in *Nrxn3* KO neurons prompted us to assess the nanoscale properties of PSD-95, another key constituent of synaptic nanocolumns[3]. We observed that the single molecule localizations of PSD-95 formed synaptic compartments and within these compartments, PSD-95 assembled into SSDs (Fig. 2a)[3,22]. Consistent with the functional *Nrxn3* KO phenotype at excitatory synapses, we observed a 47% reduction in PSD-95 compartment volume and a 22% decrease in the number of PSD-95 SSDs per synapse (Fig. 2a–c). However, *Nrxn3* KO did not alter PSD-95 SSD volume (Fig. 2d). These nanoscopic reductions in PSD-95 are likely correlated with the GluA1 phenotype as the deficits are similar (Fig. 1b–d). However, unlike the nanoscopic properties for GluA1 (Fig. 1e), the relative density of PSD-95 localizations in SSDs was unchanged (Fig. S2a). To determine the nanoscale alignment of PSD-95 in synaptic nanocolumns, we co-labeled the neurons with RIM1. The deletion of *Nrxn3* again resulted in a significant reduction in RIM1 compartment volume and SSD volume with no change in the number of RIM1 SSDs (Fig. 2e–g). Importantly, we found no significant changes in nanocolumn alignment of PSD-95 and RIM1 densities using enrichment analysis and paired cross-correlation approaches, suggesting that although the density of GluA1 is decreased in synaptic nanocolumns after the genetic ablation of *Nrxn3*, the density of RIM1 and PSD-95 in transsynaptic nanocolumns persists (Fig. S2b–d). While the effect of *Nrxn3* KO on PSD-95 and GluA1 density opposite RIM1 differs, PSD-95 is a multifunctional scaffold and displays only partial co-localization with GluA1 at the nanoscale level[23]. The sustained transsynaptic alignment of RIM1 and PSD-95 densities provides insight into how NMDAR-mediated synaptic transmission is unaltered in *Nrxn3* KO hippocampal neurons[9] as PSD-95 directly interacts with NMDARs and is required for NMDAR surface expression and function[26–28]. Taken together, these data suggest that *Nrxn3* may perhaps contribute to the subsynaptic stabilization of RIM1 and PSD-95 via intracellular and transsynaptic signaling, respectively, but it is not required for the

nanocolumnar alignment of these proteins. By contrast, *Nrxn3* is required for both the nanoscale stabilization and transsynaptic alignment of AMPARs in hippocampus. The conditional expression of non-functional HA-Nrxn1 did not impact the nanoscale organization of RIM1 or PSD-95 (Fig. S2e–m). Together, our 3D dSTORM data provide critical nanoscale insight into how the ablation of *Nrxn3* manifests as a reduction in AMPAR-mediated synaptic strength.

### Deletion of *Neurexin-3* in primary cortical neurons does not alter GluA1 nano-organization

The nanoscale phenotype we observed with *Nrxn3* KO in hippocampal cultures prompted us to ask whether *Nrxn3* controls excitatory synapse nano-organization in other brain regions. While the functional effects of *Nrxn3* manipulation in hippocampus have been examined[8–10], it remains untested whether *Nrxn3* controls excitatory transmission in cortex. Unlike the hippocampus where Nrxn3α/β is highly expressed in excitatory principal neurons, Nrxn3α/β mRNAs in cortex are more abundant in GABAergic neurons across layers and its mRNA levels vary by cortical region and layer[29]. We co-stained for GluA1 and RIM1 and found that *Nrxn3* KO did not impact GluA1 SSD properties, RIM1 SSD properties, or synaptic nanocolumn alignment (Figs. 2h–n and S2n–p). Perhaps these findings are not surprising given that *Nrxn3* is pleiotropic and can control distinct aspects of synaptic transmission in a brain-region and cell-type-specific manner[9,13–15,30,31]. Thus, while our superresolution imaging data suggest that *Nrxn3* does not control AMPAR-mediated synaptic transmission in cortical culture, *Nrxn3* may be controlling different functional parameters at cortical synapses.

### Computational modeling indicates that the deficits in GluA1 nanoscale properties in *Nrxn3* KO neurons impairs AMPAR-mediated synaptic transmission

It is critical to determine whether the altered GluA1 nanoscale properties observed in *Nrxn3* KO hippocampal neurons can explain the functional AMPAR phenotype at hippocampal synapses in primary culture and ex vivo slices. We adopted a previously validated computational model of AMPAR-mediated synaptic transmission at hippocampal synapses that incorporated defined presynaptic release, rates of glutamate diffusion, the biophysical properties of AMPAR function as well as the experimentally determined AMPAR nanoscale parameters defined here[17,32–40]. A synapse containing a single presynaptic RIM1 SSD and postsynaptic GluA1-containing SSD was modeled, which assumes that action potential-triggered vesicle release is confined to RIM1 SSDs, which activates postsynaptic AMPARs through a 9-step kinetic process (Fig. 3a–c). To first assess the contribution of different populations of AMPARs at increasing radii from the center of a RIM1 SSD to an evoked AMPAR EPSC, we divided the synapse into three concentric rings based on our STORM measurements. The first ring contained the GluA1 SSD, which is represented by an area with a radius of 40 nm from the center of the RIM1 SSD and contains the peak density of GluA1. The second ring formed an annulus with an inner

## Neurexin-3 knockout impairs PSD-95 nano-organization in hippocampus

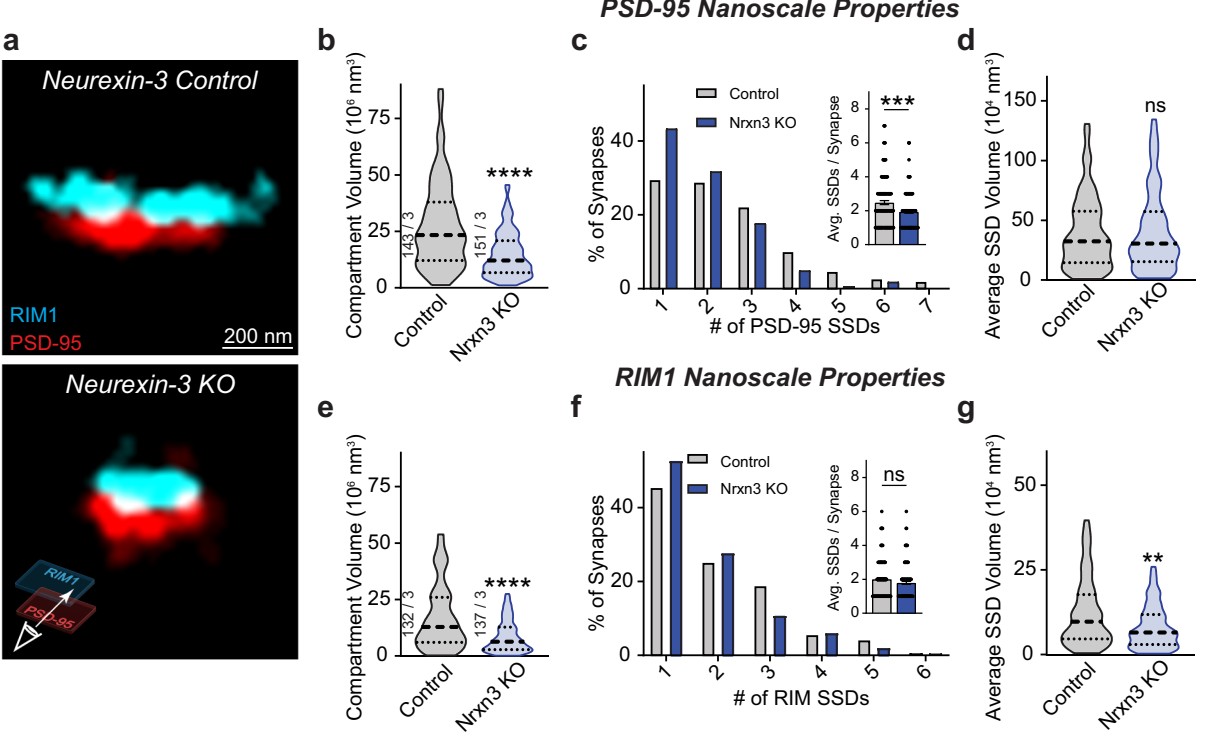

### PSD-95 Nanoscale Properties

### RIM1 Nanoscale Properties

## Neurexin-3 knockout in cortical cultures does not alter GluA1/RIM1 properties

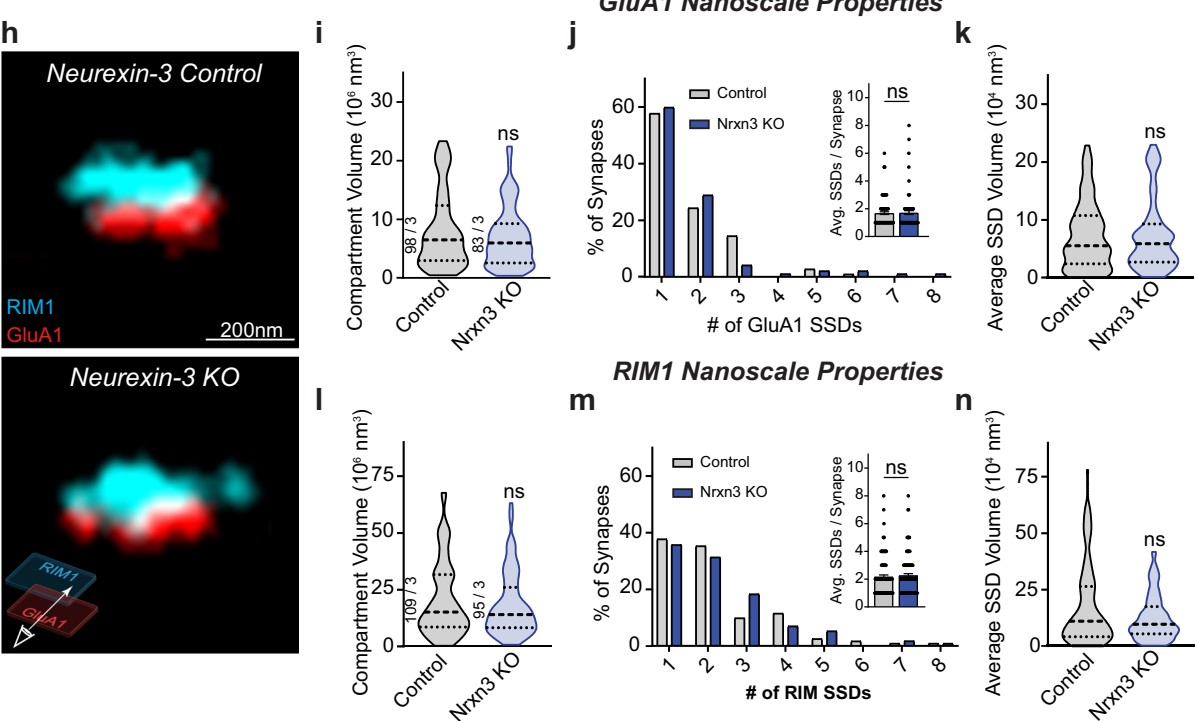

### GluA1 Nanoscale Properties

### RIM1 Nanoscale Properties

radius of 40 nm and an outer radius of 80 nm corresponding to the radius at which GluA1 density approached a randomized distribution (Fig. 1j), which we termed the peri-SSD region. The third ring formed an annulus with an inner radius of 80 nm and an outer radius of 250 nm corresponding to the approximate radius of PSD-95 in control synapses, which we termed the peripheral or out-of-SSD region (Fig. 3d). We then populated AMPARs into each ring proportional to our experimentally determined AMPAR densities for control hippocampal synapses (AMPARs per ring: Ring 1: 4; Ring 2: 6; Ring 3: 3) and simulated the AMPAR EPSC amplitudes for each ring while taking into account AMPAR activation kinetics and the rate of glutamate diffusion. We found that AMPARs in rings 1 and 2, which correspond to the SSD and

**Fig. 2 | Ablation of *Nrxn3* impairs the nano-organization of PSD-95 in hippocampus. a** Reconstructed 3D dSTORM images of presynaptic RIM1 and postsynaptic PSD-95 in control (top) and *Nrxn3* KO hippocampal neurons (bottom). *Nrxn3* KO reduces PSD-95 compartment volume $p < 0.0001$ (**b**) and SSDs per synapse $p = 0.0005$ $n = 149$ control and $n = 164$ *Nrxn3* KO synapses (**c**) without changing SSD volume $p = 0.8609$ $n = 123$ control and $n = 136$ *Nrxn3* KO synapses (**d**). Significance: Mann–Whitney test (two-tailed). **e–g** Same as in (**b–d**) but for RIM1. RIM1 compartment volume, $p < 0.0001$; SSDs per synapse, $p = 0.1097$ $n = 143$ control and $n = 148$ *Nrxn3* KO synapses; SSD volume, $p = 0.0017$ $n = 115$ control and 126 *Nrxn3* KO synapses. Significance: Mann–Whitney test (two-tailed). **h** Reconstructed 3D dSTORM images of presynaptic RIM1 and postsynaptic surface GluA1 in control (top) and *Nrxn3* KO cortical neurons (bottom). *Nrxn3* KO does not alter nanoscale GluA1 properties including compartment volume $p = 0.4802$ (**i**), SSDs per synapse $p = 0.6117$ $n = 111$ control and $n = 97$ *Nrxn3* KO synapses (**j**), or SSD volume $p = 0.8620$ $n = 85$ control and $n = 72$ *Nrxn3* KO synapses (**k**). Significance: Mann–Whitney test (two-tailed). *Nrxn3* KO results in no changes in nanoscale RIM1 properties including compartment volume $p = 0.5918$ (**l**), SSDs per synapse $p = 0.5806$ $n = 122$ control and $n = 115$ *Nrxn3* KO synapses (**m**), or SSD volume $p = 0.3018$ $n = 100$ control and $n = 96$ *Nrxn3* KO synapses (**n**). Significance: Mann–Whitney test (two-tailed). Data from three independent experiments. Number of synapses indicated on the graph unless stated in the legend. Bar graphs and line graphs: average ± SEM. Violin plots: median ± upper and lower quartiles. $**p < 0.01$; $***p < 0.001$; $****p < 0.0001$. Source data are provided as a Source Data file.

in peri-SSD regions, significantly contributed to the overall AMPAR EPSC amplitude (Fig. 3e). By contrast, peripheral AMPARs in ring 3 contributed <10% of the overall EPSC amplitude (Fig. 3e). The overall simulated AMPAR EPSC amplitude modeled the contribution of AMPAR densities in rings 1–3. The computationally modeled results are consistent with previous simulations suggesting the density of AMPARs immediately adjacent to neurotransmitter release sites is critical for synaptic transmission[5].

We next simulated evoked AMPAR-mediated current at control synapses and then sequentially introduced three GluA1 nanoscale parameters impacted in the *Nrxn3* KO, as measured by 3D dSTORM, to the model (Table 1). First, we modeled the effect of a 17.4% reduction in GluA1 SSDs per synapse. Our model stimulates a single synapse containing one pre- and postsynaptic SSD, thus, to account for the reduction in the number of SSDs per synapse in the *Nrxn3* KO, the simulated AMPAR EPSC amplitudes were multiplied 10 times for control and 8.2 times for the *Nrxn3* KO. The 17.4% reduction in GluA1 SSDs decreased the AMPAR EPSC amplitude by 18% (Fig. 3f). Second, we modeled how structural changes to the radii of the compartment and SSD were impacted. We measured a non-significant increase in the radius of GluA1 SSDs and a significant decrease in the radius of the GluA1 compartment. Factoring these parameters into the model, we found that AMPAR EPSC amplitude was increased by 6% (Fig. 3f). Third, we modeled the 33% reduction in GluA1 density opposite RIM 1 SSDs in *Nrxn3* KO synapses, represented by a decrease in GluA1 density in rings 1 and 2 (Fig. 1j). Reduced GluA1 localization density in SSDs resulted in a 24.4% decrease in AMPAR EPSC amplitude (Fig. 3f). We computationally modeled all three parameters affected in *Nrxn3* KO neurons (SSD numbers per synapse, structural changes and GluA1 density), and AMPAR EPSC amplitudes were reduced by 36.3%. The 29% increase in GluA1 density within the postsynaptic compartment (Fig. S1c) appeared paradoxical considering the 33% decrease in GluA1 density opposite RIM1 and the 19.4% reduction in GluA1 molecules in SSDs. However, computational modeling suggests that this apparent increase in GluA1 density in the synaptic compartment was a result of the significant (32%) reduction in compartment volume and likely not due to the relocation of GluA1 single molecule localizations out of SSDs (Fig. 3f). Importantly, the computationally modeled 36.3% reduction is similar to the experimentally determined 40–50% reduction in AMPAR EPSC amplitudes electrophysiologically monitored from cultured *Nrxn3* KO hippocampal neurons[9] (Fig. 3f). Thus, it is likely that the direct changes in GluA1 nanoscale properties significantly contribute to the functional AMPAR phenotype, however, we cannot exclude other mechanisms not measured here that may also contribute to impaired AMPAR-mediated synaptic transmission in *Nrxn3* KO hippocampal neurons.

## Generation of a *V5-Nrxn3* mouse allows for the reliable detection of endogenous Nrxn3

Although Nrxns were discovered 30 years ago[41], the lack of reliable antibodies to individual Nrxns has left many fundamental questions unanswered. To circumvent this limitation for Nrxn3, we used CRISPR-Cas9 genome editing to insert a 42-nucleotide sequence encoding a V5 epitope tag into the constitutively utilized exon 25c[42]. Importantly, the V5 tag is in the identical position as the HA tag in the HA-Nrxn1 mouse[25] (Figs. S1f and 4a). We verified the in-frame insertion of the V5 tag, confirmed that the V5-Nrxn3 knockin does not alter survival and validated the endogenous expression of full-length V5-Nrxn3 as well as the expression of key synaptic proteins (Figs. 4b–f, S3a). Given the robust functional excitatory synaptic phenotypes observed in hippocampal neurons following the manipulation of Nrxn3, we assessed miniature excitatory postsynaptic currents (mEPSCs) in ex vivo subiculum slices[43,44]. Relative to wild-type littermates, mEPSC frequency and amplitude were unaltered in V5-Nrxn3 slices (Fig. 4g–j). We next assessed the surface expression of V5-Nrxn3 and its co-localization with the excitatory presynaptic marker, vGluT1 by confocal microscopy. We only detected anti-V5 surface immunoreactivity on V5-Nrxn3 KI neurons, which co-localized with 72% of excitatory synapses (Fig. 4k, l). To determine if V5-Nrxn3 alters the nanoscale organization of Nrxns, we performed 3D dSTORM on wild-type and V5-Nrxn3 primary neurons co-labeled with pan-Nrxn and Homer1 and found no significant differences in the pan-Nrxn nanoscale properties (Fig. S3b–d). We next examined the nanoscale architecture of RIM1 and PSD-95 in wild-type and V5-Nrxn3 primary hippocampal cultures and found no differences in the nano-organizational properties of excitatory synapses (Fig. S4a–e). These validation experiments demonstrate that the endogenous expression of V5-Nrxn3 is a viable tool to investigate the nanoscale organization and architecture of endogenous Nrxn3.

## Endogenous Neurexin-3 forms presynaptic SSDs that localize within synaptic nanocolumns in hippocampal neurons

The striking functional[9,14] and nanoscopic phenotypes resulting from the ablation of *Nrxn3* in hippocampus (Fig. 1), next prompted us to ask, what are the nanoscale properties of Nrxn3 at excitatory synapses? To address this question, we live surface labeled *V5-Nrxn3* neurons for V5 and co-stained for Homer1. Homer1 is a marker of excitatory synapses that, unlike PSD-95, exhibits a relatively homogenous distribution and provides a reliable origin for excitatory synapses from which a variety of other synaptic proteins have been measured[25,45]. 3D dSTORM of these immunolabeled neurons revealed that V5-Nrxn3 localizations organize into an average of 2 SSDs per synapse and occupy ~85% of Homer1+ synapses (Figs. 5a, b and S4f–h). Approximately half of these excitatory synapses contained a single V5-Nrxn3 SSD (Fig. 5b). To examine the radial distribution of Nrxn3 in excitatory presynapses, the relative frequency histograms of V5-Nrxn3 localizations and SSDs relative to Homer1 centroid were fit with a Gaussian distribution, which revealed the mean radial distribution of V5-Nrxn3 single molecule localizations and SSDs was ~187 nm (95% CI: 178–197) and ~186 nm (95% CI: 173–198 nm) from the center of Homer1, respectively (Fig. 5c, d).

While Homer1 is an excellent target for determining the relative position of a protein of interest at excitatory synapses, it is not considered a component of the transsynaptic nanocolumn. We therefore next co-stained for V5-Nrxn3 and PSD-95 to determine if

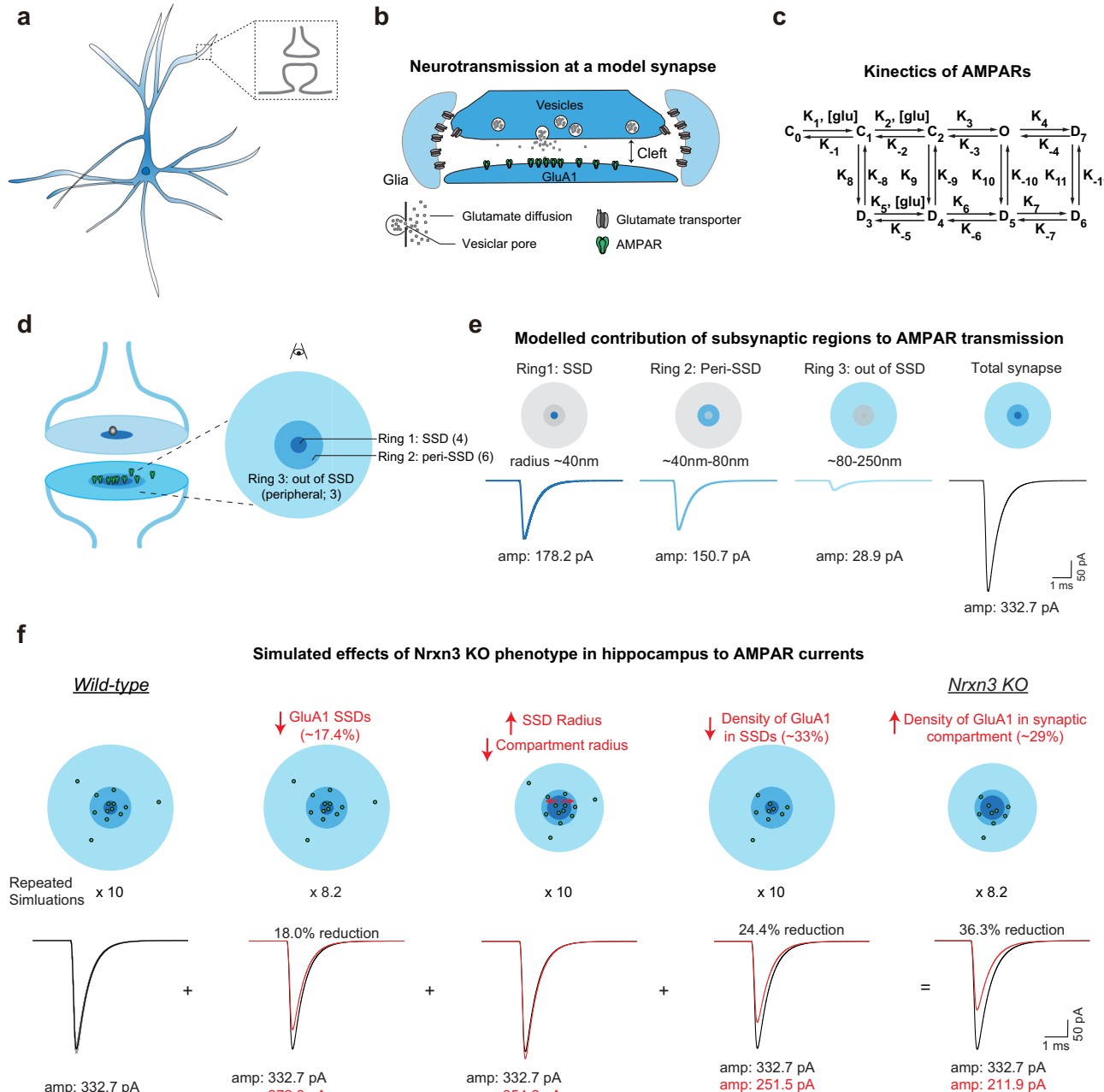

**Fig. 3 | Computational modeling of *Nrxn3* conditional knockout hippocampal synapses shows deficits in AMPAR transmission. a** Schematic of pyramidal neuron and synapse. **b** Diagram of modeled synapse. **c** Schematic of AMPAR kinetics modeled in simulation. **d** Diagram of synapse indicating regions of AMPARs included in the simulation. **e** Isolation of the AMPAR current from three regions of the modeled synapse from left to right: Ring 1 which corresponds to GluA1 SSDs, Ring 2 the peri-SSD region, and Ring 3 which is peripheral to SSDs. **f** Computational simulation of potential effects on EPSC in *Nrxn3* KO hippocampal neurons. From

left to right, there are cartoons (above) and simulated EPSC traces (below) from WT (black), decreased GluA1 SSDs (red), altered SSD and compartment volumes (red), reduced density of GluA1 (red), and the cumulative effects after *Nrxn3* KO (red). The gray traces in the left panel are 160 runs with release sites randomly distributed through the active zone, the black trace is the mean value. For parameters used, see Table 1. Schematic drawing in (**b**) is adapted from Han et al., 2022 (17). Source data are provided as a Source Data file.

V5-Nrxn3 resides in synaptic nanocolumns. 3D dSTORM of V5-Nrxn3 neurons stained for surface V5 and intracellular PSD-95 indicated that Nrxn3 is present at ~89% of PSD-95+ synapses (Fig. 5e, f; single synapse views in S4i) and that most (63%) *Nrxn3* SSDs were transsynaptically opposed to regions of significant PSD-95 density (Fig. 5g: arrow; Fig. S4j). To calculate transsynaptic density enrichment, it is necessary to first computationally align the pre- and post-synaptic compartments transsynaptically. A critical assumption for alignment is that the localizations of both pre- and post-synaptic proteins form rough disc shapes that represent the active zone or postsynaptic

density[19]. While the localizations of Nrxn3 were highly clustered, there remained enough localizations outside of SSDs to meet this criterion, which allowed for the alignment of V5-Nrxn3 and PSD-95. We calculated a median 78% transsynaptic enrichment index, indicating that we detected high-densities of PSD95 localizations opposite V5-Nrxn3 SSDs (Figs. 5h and S4k). As an alternative approach to determine if V5-Nrxn3 SSDs reside in PSD-95 nanocolumns, we quantified the nearest neighbor distances of V5-Nrxn3 and PSD-95 SSDs. Unlike the enrichment analysis, the nearest-neighbor measurements require neither synaptic localizations to form a disc-like

**Table 1 | Parameters used for computational modeling of AMPAR EPSCs in Control and *Nrxn3* KO synapses**

| Default parameters used for simulation | | |
|---|---|---|
| Time step size[31] | 0.5 µs | |
| Cleft height[32] | 15 nm | |
| Vesicle fusion duration[33,34] | 0.2 ms | |
| GluA1 number[32] | 134 | |
| Glutamate diffusion rate[32] | 0.4 µm$^2$/ms | |
| Conductance[35] | 31 pS | |
| Distance between synapse and glial sheath[31] | 40 nm | |
| The radius of the release zone | 36 nm | |
| Glutamate per vesicle | 3000 | |
| Transporter Density | 2000/µm$^2$ | |
| Parameters used in Fig. 3 (Based on STORM data) | | |
| | **WT** | ***Nrxn3* KO** |
| Radius of SSD | 0 ~ 34 nm | 0 ~ 36 nm |
| No. of GluA1 in SSD[a] | 41.6 | 31.2 |
| The radius of the synapse (peri-SSD) | 34 ~ 93 nm | 36 ~ 83.5 nm |
| No. of GluA1 in the synapse (peri-SSD)[a] | 58.4 | 54.8 |
| Radius out of SSD | 93 ~ 273 nm | 83.5 ~ 196 nm |
| No. of GluA1 out of SSD | 34 | 3 |

[a]The number of receptors does not need to be integers because the simulation can be regarded as an average of a series of simulation runs with integral receptor numbers.

shape nor computational transsynaptic alignment. The median nearest neighbor distance was ~119 nm for Nrxn3 to PSD-95 and ~146 nm for PSD-95 to Nrxn3 (Fig. 5i, j inset). We compared these experimentally defined distributions to a randomized control and found the experimental nearest neighbor distances were closer than predicted by chance (Fig. 5i, j). Thus, these data indicate that Nrxn3 is present at most excitatory synapses and assembles into SSDs near transsynaptic nanocolumns.

**Endogenous Neurexin-1 localizes to the periphery of excitatory synapses and synaptic nanocolumns**

Nrxn1 and Nrxn3 control distinct and non-overlapping functional properties, which raises the intriguing possibility that the nanoscale distribution of Nrxn1 is distinct from Nrxn3[43]. To assess the nanoscopic properties of Nrxn1, we analyzed primary hippocampal cultures from *HA-Nrxn1* mice (Fig. S1f). While the *HA-Nrxn1* mouse has been previously studied[25], the properties and nanocolumn alignment of RIM1 and PSD-95 were not assessed. We first used 3D dSTORM to test whether the incorporation of the extracellular HA tag altered the organization of RIM1 and PSD-95. The nanoscale properties and alignment of RIM1 and PSD-95 SSDs were nearly identical between wild-type and *HA-Nrxn1* KI neurons (Fig. S5a–e). Thus, the extracellular HA tag likely does not disrupt the nanoscale architecture of excitatory synapses, which permits further interrogation into the super-resolution properties of Nrxn1. *HA-Nrxn1* neurons live surface stained for HA and for intracellular Homer1 revealed that Nrxn1 is present at 91.8% of excitatory synapses. We determined an average of ~2 Nrxn1 SSDs per synapse, however, ~50% of excitatory synapses contain one Nrxn1 SSD (Figs. 6a, b, S5f–h). Distinct from Nrxn3, which was localized ~186 nm from the centroid of the synapse (Fig. 5c, d), the radial distributions HA-Nrxn1 single molecule localizations and SSDs were localized 290 nm (95% CI: 289–292 nm) and 275 nm (95% CI: 259–291 nm) from Homer1 centroid, respectively (Figs. 6c, d, S5f). Thus, endogenous HA-Nrxn1 SSDs are found at most excitatory synapses and are localized ~90 nm more peripherally than V5-Nrxn3.

We next asked whether HA-Nrxn1 SSDs are transsynaptically localized near PSD-95 nanocolumns. 3D dSTORM imaging revealed the presence of endogenous HA-Nrxn1 localizations at the majority of PSD95 positive excitatory synapses (~94%; Figs. 6e–g and S5i). We first utilized protein density enrichment analysis to determine the normalized density of PSD-95 opposite HA-Nrxn1. Similar to V5-Nrxn3, there were sufficient HA-Nrxn1 localizations outside of SSDs to perform transsynaptic alignment and quantify the density of PSD-95 opposite HA-Nrxn1 SSDs. We found a median enrichment index of 54% (Figs. 6h and S5k). By contrast, the enrichment index was 44% greater for Nrxn3 SSDs (78%) (Fig. S4k). Given the disparate radial distributions determined for Nrxn1 and Nrxn3, we unexpectedly found that, similar to Nrxn3, PSD-95 also opposes ~60% of HA-Nrxn1 SSDs (Fig. S5j). However, the nearest neighbor distance of a Nrxn1 SSD to a PSD-95 SSD was not closer than expected by chance indicating that Nrxn1 is not tightly localized opposite to PSD-95 SSDs (Fig. 6i). The median nearest neighbor distance of PSD-95 SSDs to Nrxn1 SSDs was slightly shorter than expected by chance (11.4%), however, it was 44% farther than the nearest neighbor distance of PSD-95 to Nrxn3 SSDs (215 nm vs 149 nm, respectively) (Figs. 5j, 6j). Thus, Nrxn1 SSDs may be localized near the lower density peripheral edges of PSD-95 SSDs, which raises the appealing possibility that Nrxn1 and Nrxn3 exhibit different subsynaptic positions to enable precise transsynaptic interactions with different postsynaptic ligands.

**Endogenous LRRTM2 and GluD1 are localized to distinct postsynaptic regions**

At excitatory hippocampal synapses, Nrxn1 SS4+ may participate in a tripartite complex with cerebellin-2 (Cbln2) and GluD1 to stabilize NMDARs[43]. By contrast, Nrxn3 SS4− likely interacts with LRRTM2 to promote the surface stabilization of AMPARs[8]. To begin to decipher the logic underlying how individual Nrxns take part in specific transsynaptic interactions, we examined the subsynaptic properties of endogenous LRRTM2 and GluD1. Knockdown of endogenous LRRTM2 with a published shRNA confirmed the specificity of an extracellular N-terminal LRRTM2 antibody (Fig. S6a–c)[46]. We labeled surface LRRTM2 in primary hippocampal cultures and co-stained for intracellular Homer1. 3D dSTORM revealed that LRRTM2 is present at ~70% of Homer1+ synapses and primarily assembles into a single SSD (Fig. 7a, b). The mean radial distribution of LRRTM2 localizations and SSDs was ~184 nm from the centroid of Homer1 (localizations: 185.4 nm) (95% CI: 182.5–188.2 nm); SSDs: 183.8 nm (95% CI: 164.4–201 nm) (Fig. 7c, d), which is in agreement with the subsynaptic localization of overexpressed LRRTM2[47]. The nanoscopic radial distribution of LRRTM2 is ideally positioned to transsynaptically oppose Nrxn3 SSDs (Fig. 5d). Intrigued with the similar nanoscale distribution of Nrxn3 and LRRTM2, we next asked whether Nrxn1 or Nrxn3 SSDs are closer in proximity to LRRTM2. Importantly, the nanoscale properties of LRRTM2 were almost identical in HA-Nrxn1 and V5-Nrxn3 cultures (Fig. S6c–e). In cultured HA-Nrxn1 or V5-Nrxn3 neurons, we performed live surface co-labeling with anti-LRRTM2 and anti-HA or anti-V5 antibodies followed by permeabilized immunostaining for Homer1. Using Homer1 as a widefield mask to unambiguously identify excitatory synapses, we found that the nearest neighbor distances between LRRTM2-Nrxn3 SSDs were ~20% shorter than the nearest neighbor distances between LRRTM2-Nrxn1 SSDs (Fig. 7e, f).

Next, we determined the nanoscale localization of endogenous GluD1 in hippocampal neurons. We used a GluD1 antibody that detects cytoplasmic sequences, which differs from the extracellular antibodies used to detect HA-Nrxn1, V5-Nrxn3 and LRRTM2. Additionally, all primary antibodies used thus far for quantification were directly conjugated due to the resolution of Double Helix 3D-STORM and to avoid the possibility of antigen-fluorophore linkage errors. Thus, we first compared a knockout-validated GluD1 antibody (anti-GluD1[895-932]) that is not amenable to direct conjugation with a commercially available

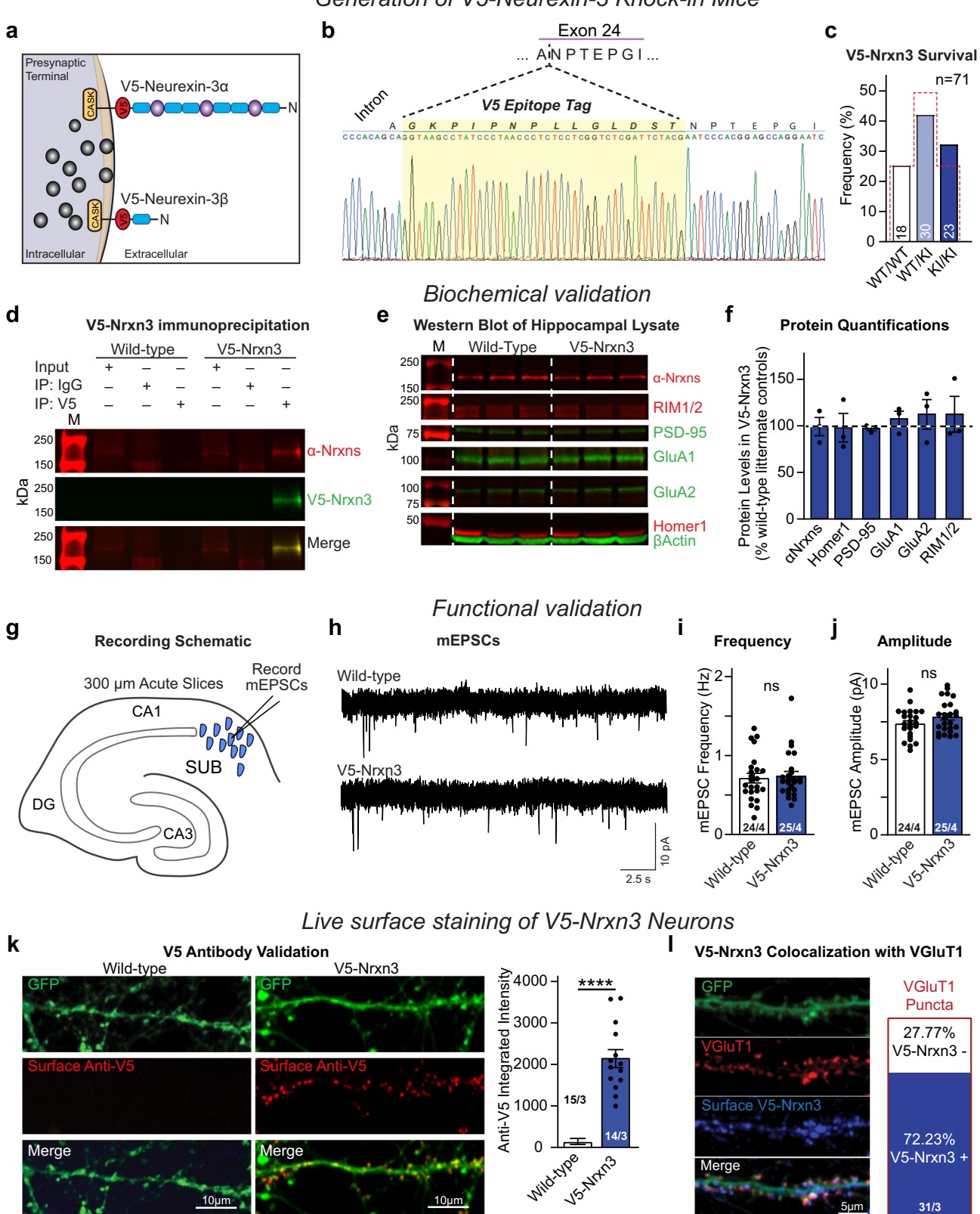

*Generation of V5-Neurexin-3 Knock-in Mice*

*Biochemical validation*

*Functional validation*

*Live surface staining of V5-Nrxn3 Neurons*

and directly conjugated GluD1 (anti-GluD1[501-600]) (Fig. S6f–j)[48]. We fixed and permeabilized neurons and co-immunolabeled them for GluD1 and Homer1. Although each antibody recognizes different intracellular epitopes on GluD1, 3D dSTORM of both antibodies revealed nearly identical nanoscale properties for all parameters measured. Both antibodies detected GluD1 at ~76% of excitatory synapses and 60–70%

of synapses contained a single SSD. Importantly, both antibodies displayed similar radial distributions of GluD1 localizations and SSDs. Direct labeling with anti-GluD1[501-600] identified GluD1 localizations and SSDs exhibited mean distributions of 222 nm (95% CI: 220.0–224.8) and 231 nm (95% CI: 215.4–245.8 nm) from Homer1 centroid, respectively (Fig. S6i–j). Indirect labeling with anti-GluD1[895-932] revealed that

**Fig. 4 | Homozygous *V5-Nrxn3* knock-in mice show no deficits in survival, synaptic protein expression, or synaptic function and are suitable for Nrxn3 localization. a** Schematic of the V5-tag knock-in site. **b** Sanger Sequencing validation of F1 *V5-Nrxn3* mice. **c** Survival rates of offspring from heterozygous *V5-Nrxn3* mice (expected Mendelian ratio = red outline) $p = 0.2999$ χ² test, $n = 71$ mice. **d–f** Immunoprecipitation of V5-Nrxn3 from whole brain lysate (**d**). Representative Western blots of excitatory synaptic proteins from wildtype littermate (left) and *V5-Nrxn3* (right) mice (**e**). Western blot quantification of synaptic proteins in *V5-Nrxn3* mice (**f**) α-Nrxns, $p = 0.5456$; Homer1, $p = 0.9076$; PSD-95, $p = 0.3289$; GluA1, $p = 0.4543$; GluA2, $p = 0.5144$; or RIM1/2, $p = 0.5853$. One sample $t$ test (two-tailed), $n = 3$ independent animals for each condition. M: molecular weight ladder.

**g–j** Functional analysis of *V5-Nrxn3* mice. Recording schematic (**g**). Representative mEPSCs from *V5-Nrxn3* and wild-type littermate controls (**h**). mEPSC frequency (**i**); ($p = 0.7493$). mEPSC amplitude (**j**); ($p = 0.1275$). Wildtype: 24 cells from 4 independent animals; *V5-Nrxn3*: 25 cells from 4 independent animals. Significance: unpaired $t$ test (two-tailed). **k** Representative confocal images (left) of surface anti-V5 labeling of wildtype and *V5-Nrxn3* neurons. Quantification (right) of surface expression of V5-Nrxn3; $n = 15$ Wild-type and $n = 14$ *V5-Nrxn3* cells from 3 independent experiments unpaired t-test (two-tailed). **l** Representative image (left) and summary graphs of V5-Nrxn3 co-localization with vGluT1 (right). Number of neurons and independent experiments are indicated in the figure. Bar graphs indicate mean ± SEM. ****$p < 0.0001$. Source data are provided as a Source Data file.

the mean distribution of GluD1 localizations was 248 nm (95% CI: 246.2–249.4) and SSDs was 244 nm (95% CI: 230.6–257.3) (Fig. S6i–j). Both antibodies indicate a peripheral nanoscale distribution of GluD1, which is consistent with previous reports[49–51]. However, the use of an intracellular antibody does not distinguish between single-molecule localizations representing surface or intracellular GluD1, and may likely underestimate the actual radial distances of the extracellular sequences from Homer1 that mediate the tripartite GluD1-Cbln-Nrxn complex. We next asked whether Nrxn1 or Nrxn3 are assembled into high density regions opposite GluD1. Similar to LRRTM2, the nanoscopic properties of GluD1 were nearly identical in cultures where HA-Nrxn1 or V5-Nrxn3 were endogenously expressed (Fig. S6k–m). Given the striking similarities of both antibodies, we surface labeled neurons to detect HA-Nrxn1 or V5-Nrxn3 and used the directly conjugated antibody to label GluD1 and co-labeled for Homer1 to identify excitatory synapses. Perhaps not surprising given the radial distribution of Nrxn1, the nearest neighbor distances for GluD1-Nrxn1 SSDs were 26.2% closer than for GluD1-Nrxn3 SSDs (Fig. 7k, l). While we are not able to definitively determine whether the detected GluD1 is surface exposed, these data are an important step toward determining the potential interacting partners of Nrxn1. Overall, these data are consistent with the proposed mechanisms underlying Nrxn1 and Nrxn3 function[8,12,43] and indicate that Nrxn1 and Nrxn3 may organize into discrete high-density SSDs near GluD1 and LRRTM2, respectively.

## Nrxn1 and Nrxn3 form discrete and nonoverlapping subsynaptic densities at excitatory synapses in hippocampal and cortical neurons

Our data raise the intriguing possibility that the spatial nanoscale segregation of Nrxn1 and Nrxn3 may be positioned to engage in parallel transsynaptic signaling pathways through GluD1 and LRRTM2, respectively. To directly assess the nanoscale properties of each Nrxn, we generated homozygous *HA-Nrxn1::V5-Nrxn3* mice (Fig. 8a). As an important control, we compared the nanoscale properties of RIM1 and PSD-95 in *HA-Nrxn1::V5-Nrxn3* neurons relative to wild-type neurons. Similar to neurons only endogenously expressing HA-Nrxn1 or V5-Nrxn3, the number of PSD-95 and RIM1 SSDs and their nanocolumn alignment in *HA-Nrxn1::V5-Nrxn3* neurons were indistinguishable from wild-type (Fig. S7a–e). We next sought to examine if HA-Nrxn1 or V5-Nrxn3, when simultaneously expressed in the same neuron, localize near RIM1 at excitatory synapses. *HA-Nrxn1::V5-Nrxn3* neurons were surface labeled for HA-Nrxn1 or V5-Nrxn3 and co-labeled for RIM1 and Homer1. Homer1 was used as a fluorescent mask to identify excitatory synapses (Fig. 8b, c). Consistent with our PSD-95 nearest neighbor data (Figs. 5h, i; 6h, i), the distance from RIM1 SSDs to the nearest Nrxn3 SSD (and vice versa) was 34.3% shorter than those determined for Nrxn1 (Fig. 8d–e). We next assessed the single molecule localizations of each Nrxn in the same synapse using dual surface labeling of *HA-Nrxn1::V5-Nrxn3* hippocampal neurons with HA-AF647 and V5-CF568 conjugated primary antibodies. We then permeabilized these neurons and performed indirect staining of Homer1 with AF488 to use as a widefield overlay to identify excitatory synapses (Fig. 8f). Consistent with individual HA-Nrxn1 and V5-Nrxn3 staining, most synapses positive for

Nrxns contained both Nrxn1 and Nrxn3 localizations (Fig. 8g). We determined that the median nearest neighbor distances of Nrxn1 SSDs to Nrxn3 SSDs was ~166 nm and Nrxn3 SSDs to Nrxn1 SSDs was ~126 nm, which did not deviate from chance, indicating that Nrxn1 and Nrxn3 SSDs are not colocalized (Fig. S7f–h).

Next, we used two complementary methods to assess the spatial relationship of HA-Nrxn1 and V5-Nrxn3 SSDs. First, for a single excitatory synapse, we generated masks for each HA-Nrxn1 and V5-Nrxn3 SSD, calculated the volume of each mask and determined the total mask intersection volume. The intersection volume is then divided by the sum of both mask volumes to determine the percent SSD overlap (Fig. 8h). This method is likely to overestimate overlap volume because the generated mask could be larger than the SSD itself (e.g. masking over a region that contains no or very few localizations) and increases the likelihood that small areas of overlap or overlap in regions of low-density are detected. We binned the resulting overlap data into 10% intervals and found ~88% of synapses have <10% overlap of Nrxn1 SSDs with Nrxn3 SSDs while ~73% of synapses have <10% overlap of Nrxn3 SSDs with Nrxn1 SSDs (Fig. 8i–k). We categorized the average of Nrxn1 and *Nrxn3* SSD overlap per synapse into three groups: minimal overlap (<10% overlap; ~80% of synapses), moderate overlap (>10–50% overlap; ~18% of synapses) and high overlap (≥50% overlap; ~2% of synapses) (Fig. 8l). Within the 0–10% bin, we discovered that Nrxn1 and Nrxn3 masks did not intersect at 61.3% of excitatory synapses (130 out of 212 synapses; Fig. S7i).

We also calculated the volume of overlapping HA-Nrxn1 and V5-Nrxn3 localizations. For each synapse, we first generated an alpha-Shape for one Nrxn (a complex polygon made from the localizations within an SSD) and determined the volume of localizations of the other Nrxn within the alphaShape (Fig. S7j). This overlap volume is divided by the total synaptic SSD volume to yield the percentage of SSD overlap per synapse. This method utilizes only single molecule localizations and will thus likely underestimate the overlap volume as at least four overlapping localizations are required to generate an alphaShape. We found >93% of synapses had minimal overlap (<10%) of HA-Nrxn1 SSDs with V5-Nrxn3 SSDs or V5-Nrxn3 SSDs with HA-Nrxn1 SSDs (Fig. S7k–m). Overall, our two complementary methods directly reveal that Nrxn1 and Nrxn3 generally occupy the same excitatory presynaptic terminals but their respective SSDs are largely discrete and non-overlapping, which may begin to explain how Nrxn1 and Nrxn3 control non-overlapping synaptic properties.

We next asked whether the spatial segregation of Nrxn1 and Nrxn3 is a nanoscale property unique to hippocampal neurons or reflective of a more general subsynaptic organization of these proteins. As a first step to address this question, we prepared *HA-Nrxn1::V5-Nrxn3* cortical cultures and co-stained for surface V5-Nrxn3 and HA-Nrxn1 and then fixed and permeabilized to stain for Homer1 to identify excitatory synapses. We first quantified the abundance of V5-Nrxn3 and HA-Nrxn1 in Nrxn+ excitatory synapses and found that while the majority (66.3%) of synapses contained both HA-Nrxn1 and V5-Nrxn3, there were at least double the proportion of synapses that contained only HA-Nrxn1 in cortex compared to hippocampus (30.8% cortex vs 12.8% in hippocampus), while 2.9% of synapses contained only V5-Nrxn3 (Figs. S7n

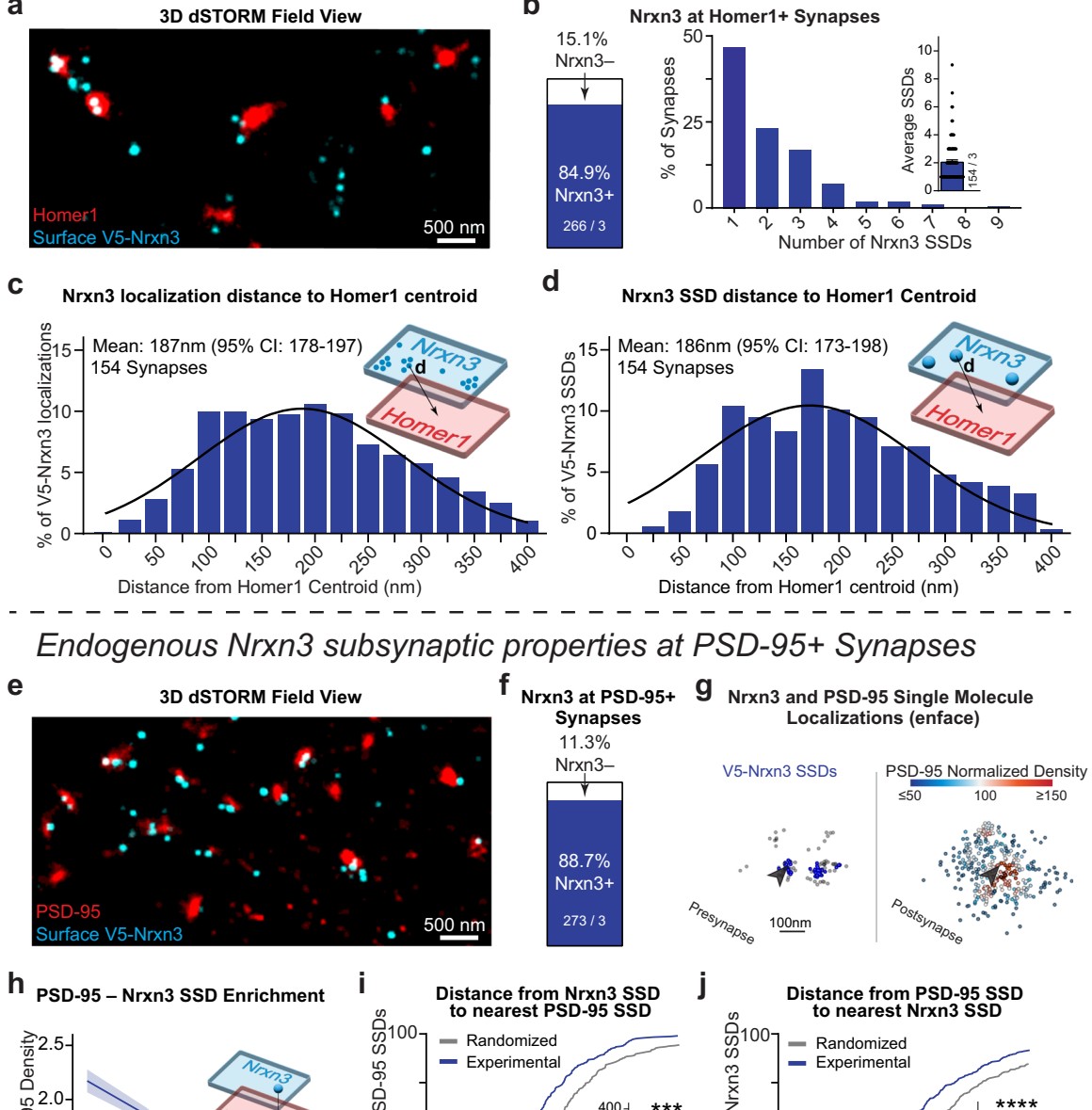

**Fig. 5 | Endogenous neurexin-3 is present at the majority of excitatory synapses and localizes near the synaptic nanocolumn. a** Representative 3D dSTORM of endogenous V5-Nrxn3 (cyan) and Homer1 (red). **b** V5-Nrxn3 Homer1 synaptic abundance (left) and number of SSDs (right). Distribution of *Nrxn3* localization distance (**c**) and SSD distance (**d**) to the centroid of Homer1. Mean and 95% confidence interval of the Gaussian fitted distribution indicated on the graph. **e** Representative 3D dSTORM of V5-Nrxn3 (cyan) and PSD-95 (red). **f** Same as (**b**) for PSD-95+ synapses. **g** Distributions of synaptic of V5-Nrxn3 (left) and PSD-95 (right). Blue spheres identify V5-Nrxn3 SSDs opposite a PSD-95 of varying density (heatmap). Arrow identifies V5-Nrxn3 SSD that opposes a region of high PSD-95 density. **h** Enrichment analysis of PSD-95 density opposite Nrxn3 SSDs. *n* = 154 synapses.

**i** Nearest neighbor distance from Nrxn3 SSDs to the closest PSD-95 SSD is left-shifted compared to randomized data. Violin plot (inset) of nearest neighbor distances, *p* = 0.0004; Mann–Whitney test (two-tailed). *n* = 303 experimental and randomized SSDs. **j** Same as in H, except for nearest neighbor distances of PSD-95 SSDs to nearest Nrxn3 SSD compared to randomized data. Violin plot (inset) of nearest neighbor distances, *p* < 0.0001; Mann–Whitney test (two-tailed). *n* = 465 experimental and randomized SSDs. Data from three independent experiments. Number of synapses indicated on the graph unless stated in the legend. Bar graphs and line graphs: average ± SEM. ***p* < 0.001; ****p* < 0.0001. Source data are provided as a Source Data file.

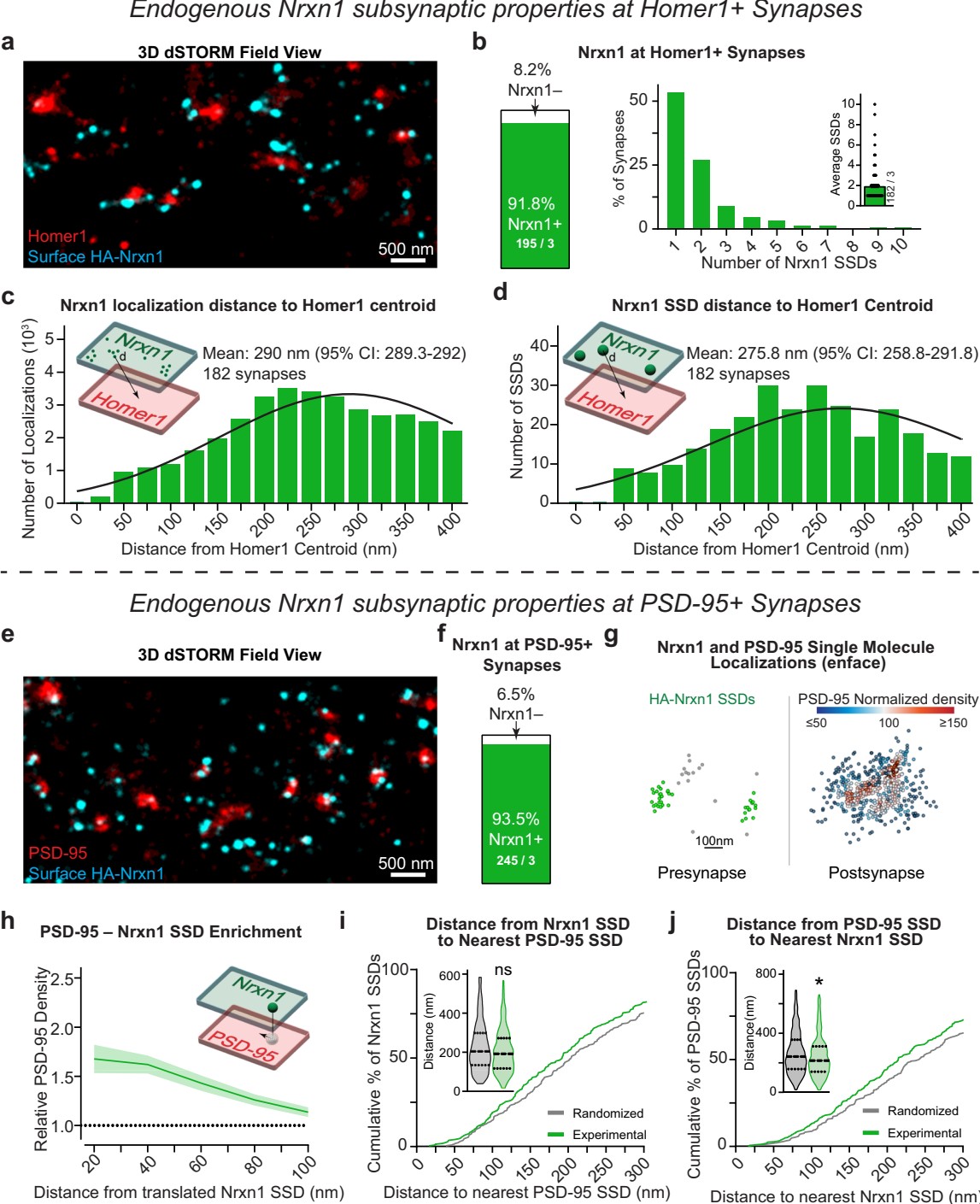

Fig. 6 | **Endogenous neurexin-1 organizes near the periphery of synapses and synaptic nanocolumns. a** Representative 3D dSTORM field view of surface HA-Nrxn1 (cyan) and Homer1 (red). **b** Nrxn1 Homer1 synaptic abundance (left) and number of SSDs (right). Distribution of Nrxn1 localization distance (**c**) and SSD distance (**d**) to the centroid of Homer1. Mean and 95% confidence interval of the Poisson fitted distribution are indicated on the graph. (**e**) Representative 3D dSTORM of Nrxn1 (cyan) and PSD-95 (red). **f** Same as (**b**) for PSD-95+ synapses. **g** Distributions of synaptic Nrxn1 and PSD-95. Nrxn1 SSDs (green spheres) opposing relative PSD-95 density (heatmap). **h** Average PSD-95 enrichment opposite Nrxn1 SSDs at increasing distances from the translated Nrxn1 SSD center.

$n$ = 111 synapses. **i** Nearest neighbor distances of Nrxn1 SSDs to PSD-95 SSDs are unchanged from randomized data. Violin plot (inset) of nearest neighbor distance, $p$ = 0.0807; Mann–Whitney test (two-tailed). $N$ = 225 experimental and randomized SSDs. **j** Same as in (**h**) but for nearest neighbor distances of PSD-95 SSDs to Nrxn1 SSD is modestly left-shifted relative to a randomized SSD location. Violin plot (inset) of median nearest neighbor distance is random, $p$ = 0.0208; Mann–Whitney test (two-tailed). $n$ = 330 experimental and randomized SSDs. Data from three independent experiments. Number of synapses indicated on the graph unless stated in the legend. Bar graphs and line graphs: average ± SEM. *$p$ < 0.05. Source data are provided as a Source Data file.

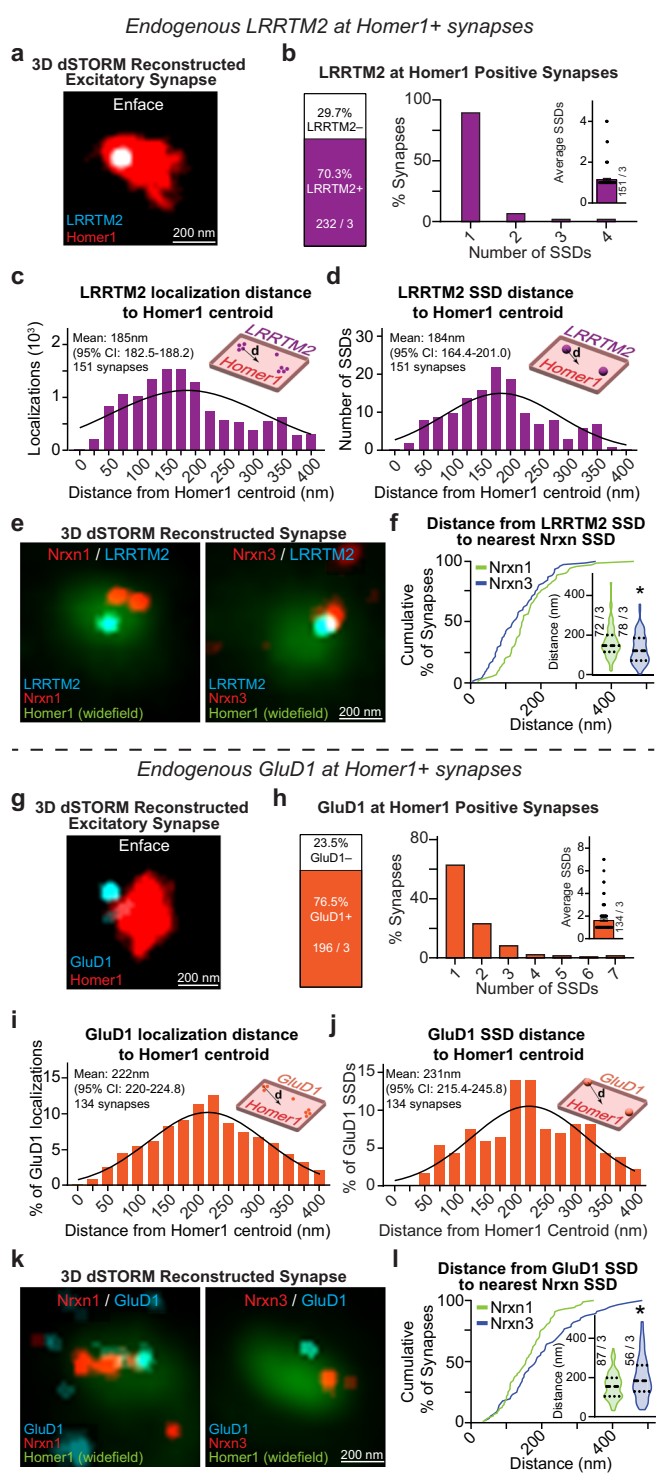

**Fig. 7 | Endogenous LRRTM2 and GluD1 assemble into SSDs near sites of neurexin-1 and neurexin-3 enrichment. a** Representative 3D dSTORM reconstruction of LRRTM2 (cyan) and Homer1 (red). **b** LRRTM2 Homer1 synaptic abundance *n* = 232 synapses from 3 independent experiments (left) and number of SSDs *n* = 151 synapses from 3 independent experiments (right). LRRTM2 localization distance (**c**) and SSD distance (**d**) to Homer1 centroid. The mean and 95% confidence interval of Poisson fitted distribution are indicated on the graph. **e** Representative 3D dSTORM reconstruction of LRRTM2 (cyan) and Nrxn1 (left) or Nrxn3 (right; red) at Homer1+ synapse (green) in an enface view. **f** Nrxn1 is farther from LRRTM2 SSDs than Nrxn3, *p* = 0.0135; Mann–Whitney test (two-tailed). **g** Representative 3D dSTORM reconstruction of GluD1 (cyan) and Homer1 (red) in an enface view. **h** GluD1 Homer1 synaptic abundance *n* = 196 synapses from 3 independent experiments (left) and number of SSDs *n* = 134 synapses from 3 independent experiments (right). **i–j** GluD1 localization distance (**g**) and GluD1 SSD distance (**h**) from the Homer1 centroid. The mean and 95% confidence interval of the Poisson fitted distribution indicated on the graph. **k** Representative 3D dSTORM reconstruction of GluD1 (cyan) and Nrxn1 (left) or Nrxn3 (right; red) at Homer1+ synapse (green) in an enface view. **l** Nrxn1 is closer to GluD1 SSDs than Nrxn1, *p* = 0.0314; Mann–Whitney test (two-tailed). Number of synapses and independent experiments are indicated in the figure. Bar graphs and line graphs: average ± SEM. Violin plots: median ± upper and lower quartiles. *\*p* < 0.05. Source data are provided as a Source Data file.

control aspects of synaptic function across disparate brain regions, circuits, and synapses.

## Discussion

We report that the genetic ablation of *Nrxn3* impairs multiple aspects of GluA1 nanoscale architecture relevant to synapse function in hippocampus – we found that there were fewer GluA1 SSDs per synapse, that the number of GluA1 molecules inside SSDs was reduced, and that the transsynaptic alignment of GluA1 localization opposite RIM1 SSDs was decreased. Further, the density of RIM1 opposite a GluA1 SSD was not impaired, indicating that, in agreement with the functional phenotype, the *Nrxn3* KO phenotype manifests postsynaptically (Fig. 9)[5,8].

Similar reorganization of AMPARs was observed after the shRNA-mediated knockdown of endogenous LRRTM2 and replacement with an exogenous LRRTM2 harboring an engineered proteolytic cleavage site[5]. Although the cleavage of exogenous LRRTM2 produced a similar nanoscale reorganization of GluA1 SSDs as those reported here, only deficits in evoked transmission were reported. This is in contrast to the functional phenotype of *Nrxn3* KO, which decreased both evoked and spontaneous mEPSCs[8,9]. This seeming discrepancy may be explained by one or more of the following possibilities: 1. Unlike genetic knockouts, shRNA-mediated knockdown suppresses, but does not eliminate, endogenous protein expression and the efficiency of suppression can be variable between cells[52]. Thus, even though the LRRTM2 shRNA used was highly efficient, endogenous LRRTM2 is likely still produced and would be insensitive to enzymatic cleavage. The remaining endogenous LRRTM2 may be sufficient to sustain mEPSCs. 2. The analysis of GluA1 nanoscale properties used overexpressed SEP-GluA1, which differs from our analyses in which we measured the nanoscopic properties of endogenous GluA1. 3. The genetic ablation of *Nrxn3* eliminates all Nrxn3-mediated transsynaptic interactions, including those with LRRTM1 and LRRTM2. The loss of multiple Nrxn3-dependent transsynaptic interactions, in addition to LRRTM2, may explain why *Nrxn3* KO impacts evoked and miniature EPSCs. In support of this notion, similar to the *Nrxn3* KO, the double knockout of LRRTM1 and LRRTM2 exhibited a decrease in both evoked and miniature AMPAR EPSCs amplitudes in hippocampus[53]. 4. Unlike *Nrxn3* KO, which reduced the total levels of endogenous surface GluA1[9] and GluA1 density in transsynaptic nanocolumns (Fig. 1j–l), the acute cleavage of LRRTM2, did not alter the surface levels of SEP-GluA1 in the synaptic compartment yet impaired GluA1 SSDs[5]. These differences

and [8]g). We next tested whether the distribution of V5-Nrxn3 SSDs and HA-Nrxn1 SSDs was nonrandom using nearest neighbor analysis and found that V5-Nrxn3 SSDs and HA-Nrxn1 SSDs were not closer together than predicted by chance (Fig. S7o, p). To directly assess overlap, we again utilized our mask method (Fig. 8h) to quantify the overlap of V5-Nrxn3 SSDs and HA-Nrxn1 SSDs in cortical cultures and found that 91.5% of synapses contained less than 10% overlap (Fig. S7q). The localization method (Fig. S7j) revealed that in ~96% of excitatory cortical synapses analyzed, less than 10% overlap was detected between HA-Nrxn1 and V5-Nrxn3 SSDs (Fig. S7r). Together these data show that the spatial segregation of HA-Nrxn1 and V5-Nrxn3 occurs across brain regions and may represent a critical property of the Nrxns to distinctly

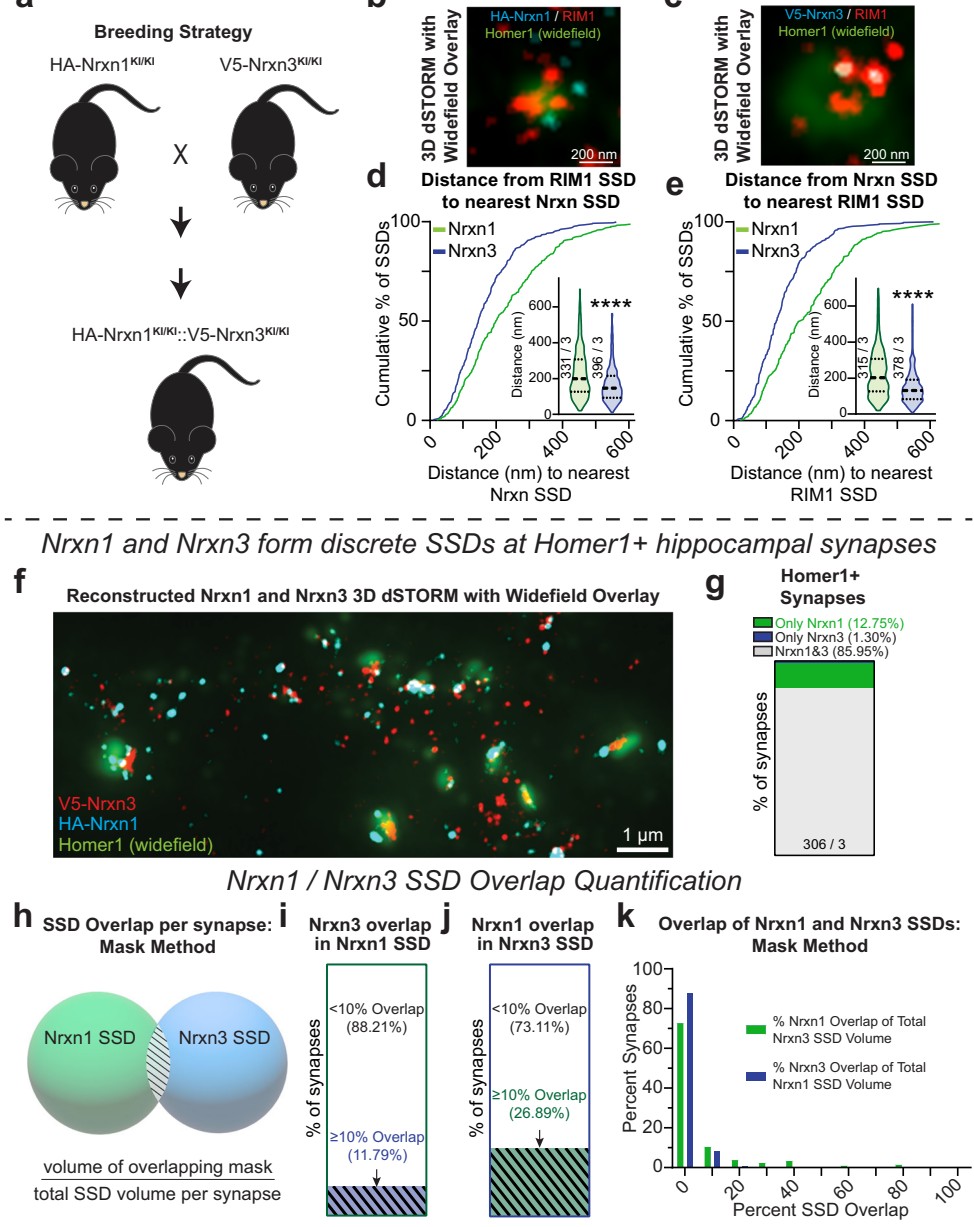

*Nrxn3 SSDs localize closer to RIM1 SSDs than Nrxn1 SSDs at Homer1+ hippocampal synapses*

*Nrxn1 and Nrxn3 form discrete SSDs at Homer1+ hippocampal synapses*

*Nrxn1 / Nrxn3 SSD Overlap Quantification*

in surface GluA1 levels after each manipulation may suggest that transsynaptic nanocolumn alignment is necessary for evoked transmission while spontaneous mEPSCs are dependent on the number of GluA1 in the synaptic compartment[54]. Addressing how the binding of different ligands to individual Nrxns governs synaptic properties is a significant priority toward understanding the combinatorial code of Nrxns at the synapse.

Unexpectedly, we also observed that the KO of *Nrxn3* reduced the volume of synaptic RIM1 as well as the volume of RIM1 SSDs in hippocampus. These presynaptic nanoscopic changes were unexpected because the ablation of *Nrxn3* does not alter presynaptic release probability. However, neither the number of RIM1 SSDs nor the relative density of molecules within each SSD were impacted, suggesting that these functional release sites are preserved. Further,

**Fig. 8 | Nrxn1 and Nrxn3 form discrete SSDs at Homer1+ hippocampal synapses.** **a** Breeding strategy to create *HA-Nrxn1^KI/KI^::V5-Nrxn3^KI/KI^* mice. Representative 3D dSTORM field view of Nrxn1 (**b**); (cyan) or Nrxn3 (**c**); (cyan) and RIM1 (red) with a widefield Homer1 overlay (green). The nearest neighbor distance is shorter from RIM1 SSDs to Nrxn3 SSDs than to Nrxn1 SSDs, *p* < 0.0001; (**d**) and from Nrxn3 SSDs to RIM1 SSDs, *p* < 0.0001; (**e**). Significance: Mann–Whitney test (two-tailed). **f** Representative 3D dSTORM field view of Nrxn3 (red) and Nrxn1 (cyan) with a wide field Homer1 overlay (green). **g** Stacked bar graph of excitatory synapses with ≥5 Nrxn1 and/or Nrxn3 localizations. **h** Schematic of the overlap mask method. SSD

overlap is determined by quantifying the volume overlap of Nrxn1 and *Nrxn3* SSDs masks. Stacked bar graph of the SSD volume overlap of Nrxn3 with Nrxn1 (**I**) and Nrxn1 with Nrxn3 (**J**). *n* = 212 synapses. (**k**) Histogram of the percent overlap of Nrxn1 and Nrxn3 SSDs using the overlapping mask method. *n* = 212 synapses. **l** Representative scatter plots depicting Nrxn1 SSDs (green) and Nrxn3 SSDs (blue) showing minimal (<10%, left), moderate (10–50% middle), and high (>50%, right) overlap of SSDs. Data from three independent experiments. Number of synapses indicated on the graph unless stated in the legend. Violin plots: median ± upper and lower quartiles. ****p* < 0.0001. Source data are provided as a Source Data file.

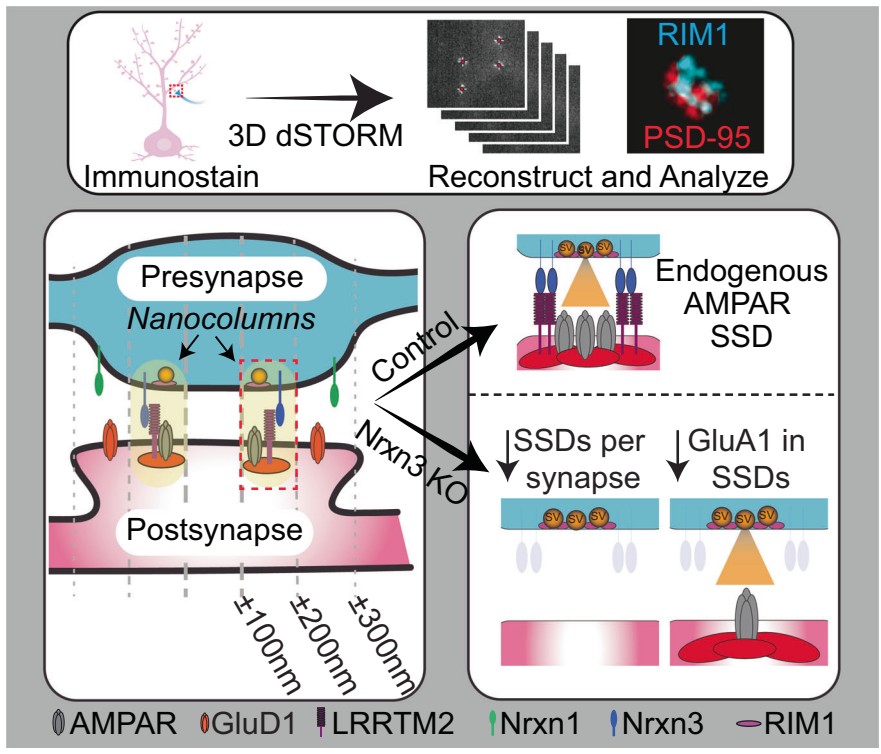

**Fig. 9 | Summary model: Nrxn3 controls excitatory synapse nano-organization in hippocampus and localizes discretely from Nrxn1.** We find that Nrxn3 controls excitatory synapse nano-organization in hippocampus and that Nrxn1 and Nrxn3

localize in discrete, non-overlapping SSDs that are preferentially opposite GluD1 and LRRTM2 SSDs respectively. Nrxn3 deletion results in decreased AMPAR SSDs per synapse as well as a decrease in the relative number of AMPARs in SSDs.

presynaptic release may be sustained by compensation with RIM2 following the deletion of Nrxn3[55]. By contrast, endogenous expression of non-functional Nrxn1 did not alter the nanoscale properties of GluA1, PSD-95 or RIM1 (Figs. S1f–t and S2e–m). In contrast to hippocampal cultures, *Nrxn3* KO in primary cortical neurons did not disrupt the nanoscale organization of excitatory synapses. To our knowledge, a functional role for Nrxn3 at excitatory synapses in cortical neurons has not been evaluated. The absence of a nanoscale phenotype at excitatory synapses in cortical culture raises three interesting possibilities for future study. First, Nrxn3 might not control AMPAR-mediated synaptic transmission in cortex or that the nanoscale mechanism by which Nrxn3 may control AMPAR currents in cortex differs from in hippocampus. Second, Nrxn3α and Nrxn3β are not highly expressed in all cortical layers[29] and our STORM data from cortical culture indicate that, unlike hippocampal culture, Nrxn3 may be absent from a significant fraction (~30%; Fig. S7n) of excitatory synapses. Thus, the relative differences in expression and localization in primary cortical neurons may obfuscate the impact of *Nrxn3* KO on nanoscale organization. Third, Nrxn3 can control distinct aspects of synaptic transmission in a brain-region specific manner, it is therefore possible that Nrxn3 may regulate synaptic properties other than AMPAR stability in cortex[9,13–15,30,31]. Overall, we identify Nrxn3 as a key functionally-relevant presynaptic adhesion

molecule responsible for organizing the nanoscale architecture of excitatory synapses in hippocampus.

We next used a validated computational model of excitatory synaptic transmission[17] to determine if the observed nanoscale changes at excitatory synapses with *Nrxn3* KO could account for the previously described functional phenotype (Fig. 3)[9]. Computational modeling of the observed changes in the overall size of the synapse, the size of GluA1 SSDs and in the density of GluA1 receptors in the synaptic compartment and SSDs resulted in a decrease in EPSCs of 36.0% (Fig. 3f). This decrease resembles the previously reported functional AMPAR phenotype of 40–50% monitored from hippocampal cultures and ex vivo slices[9]. However, we cannot exclude the contribution of other nanoscale properties not quantified here to the functional phenotype. Our STORM data are in agreement with our previous confocal measurements that detected a significant decrease in GluA1 puncta size but not intensity in *Nrxn3* KO hippocampal cultures[9]. We observed a significant decrease in the compartment volume of GluA1 (Fig. 1c) and the reorganization of GluA1 from SSDs to the synaptic compartment may account for the lack of a significant deficit in overall intensity. What accounts for the loss of GluA1 in SSDs? Our previous work suggested that the genetic ablation of *Nrxn3* increased AMPAR internalization[9], however, whether AMPAR internalization primarily affects AMPARs inside or outside of SSDs is

unknown and would require reliable photostable dyes to accommodate 3-channel dSTORM. Whether changes in the synaptic retention of GluA1 also contribute to the *Nrxn3* KO phenotype remains unclear. Previous work suggests that AMPARs that diffuse into SSDs are weakly confined and do not become incorporated into SSDs[23]. However, AMPAR SSDs do not always co-localize with PSD-95, thus, whether the SSDs that do capture diffusing AMPARs participate in functional nanocolumns remains unknown[23]. Overall, the computational modeling of the nanoscale phenotype in *Nrxn3* KO hippocampal cultures results in a decrease in AMPAR-mediated transmission that closely resembles the previously reported functional phenotype[9].

We developed a V5 epitope-tagged *Nrxn3* mouse, which permits the detection of all full-length transmembrane domain-containing Nrxn3 isoforms. The mouse generated here contrasts with a recent epitope-tagged *Nrxn3* mouse that only permits the detection of a truncated, GPI-anchored Nrxn3 SS5 variant[15]. Using our V5-Nrxn3 mouse and a previously validated *HA-Nrxn1* mouse, we characterized the subsynaptic properties of each Nrxn individually. Nrxn1 and Nrxn3 are present in the majority of excitatory synapses in hippocampus, which contrast with the initial characterization of these *HA-Nrxn1* mice that found Nrxn1 at only ~40% of excitatory synapses (Figs. 5b and 6b)[25]. Methodological differences in the acquisition of STORM data (e.g. the use of dye-pair STORM and the number of localizations collected) may account for these differences. Additionally, the presence of Nrxns at nearly all excitatory synapses provides a more direct interpretation of the large functional effects of *Nrxn1* and *Nrxn3* manipulation on NMDAR-mediated transmission and AMPAR-mediated transmission, respectively[43].

The nanoscale organization of Nrxn1 and Nrxn3 differ in their enrichment in nanocolumns and radial distributions, which supports the idea that Nrxn1 and Nrxn3 are functionally distinct (Figs. 8f–k, 9)[10]. PSD-95 is highly enriched opposite Nrxn3 SSDs but only shows a modest increased density opposite Nrxn1 SSDs. We found that Nrxn1 SSDs display a similar radial distribution with GluD1 and Nrxn3 SSDs share a similar radial distribution with LRRTM2 (Figs. 5–7 and 9). The radial positions for these Nrxn ligands are consistent with previous electron microscopy of endogenous GluD1[49,50] and super-resolution microscopy of overexpressed LRRTM2[47]. Although the radial distribution of the Nrxns and these ligands is largely overlapping, we were not able to directly test for binding of these proteins due to limitations of tools for studying endogenous protein. Neuroligin-1, a prototypical Nrxn excitatory ligand, was not included in this study because all antibodies tested exhibit non-specific binding[56]. To directly assess the nanoscale properties of endogenous Nrxn1 and Nrxn3 in the same excitatory synapse, we generated a *HA-Nrxn1::V5-Nrxn3* double KI mouse. Although both Nrxns reside in the majority of excitatory synapses, they are not in close proximity and exhibit minimal co-localization in hippocampus (Fig. 8f–l). We then performed identical sets of experiments in cortical cultures and found that Nrxn1 and Nrxn3 SSDs were similarly non-overlapping but notably detected a much larger proportion of synapses that contained only Nrxn1 in cortex (30.8%) than in hippocampus (12.8%) (Figs. S7n–r and 8g). Due to inherent biophysical differences of the dyes used to label Nrxn1 and Nrxn3, the direct comparison of single molecule localizations of Nrxn1 and Nrxn3 was not possible[57]. Despite this limitation, we demonstrate that the discrete organization of Nrxn1 and Nrxn3 SSDs may be a general nanoscale property of Nrxns and provides important insight into how individual Nrxns may govern distinct aspects of synapse function.

Recently, an elegant study used X10 expansion microscopy to characterize the nanoscopic properties of GluD1, LRRTM1 and HA-Neurolign-1[51]. Using the nanoscale localization of these ligands as well as their SS4-dependent binding properties as a correlate for neurexin SSD localization, they proposed that the key defining characteristic of Nrxn SSDs is SS4 exon usage and not the identity of individual Nrxns.

They posited that SS4+ Nrxns localize peripherally while SS4– Nrxns localize more centrally at excitatory synapses. In hippocampal neurons, Nrxn1 SS4– and Nrxn3 SS4– mRNA and protein are primarily expressed[43,58], thus, if Nrxns segregate based on SS4 identity, we would predict significant overlap of Nrxn1 and Nrxn3 within SSDs – particularly those that localize centrally in the synapse, however, we observed minimal overlap between Nrxn1 and Nrxn3 SSDs at excitatory hippocampal synapses (Figs. 8k and S7m). Our work directly interrogating the nanoscopic organization of Nrxns supports a model where individual Nrxns form distinct and non-overlapping SSDs that exhibit differential subsynaptic localization. However, the genetic approaches in our current study do not differentiate between Nrxn SS4 isoforms and thus Nrxn1 and Nrxn3 SSDs may comprise a mixture of SS4+ and SS4– isoforms or may be further segregated by the presence or absence of the SS4 insert. Our proposed model raises the intriguing possibility of parallel synaptic signaling mediated by these discrete Nrxn SSDs.

Our findings raise several critical questions about the molecular mechanisms that control Nrxn nanoscale architecture. It is unlikely that these nanoscale differences are driven exclusively by Nrxn-ligand binding affinities because relative to Nrxn3, Nrxn1 exhibits preferential binding to both Cbln2 and LRRTM2[58–60]. First, although our data suggest that Nrxn SSDs are not exclusively segregated by SS4 identity, does the inclusion or exclusion of the SS4 exon change the number and/or nanoscale localization of Nrxn SSDs? The forced expression of Nrxn3 SS4+ activates Cbln2/GluD1 signaling, suggesting that SS4 identity may redistribute Nrxn SSDs in the synapse[43]. Second, what role do intracellular binding partners and/or posttranslational modifications play to promote the homotypic assembly and nanoscale spatial segregation observed here? Validated intracellular partners such as CASK and Mint1[43] appear to indiscriminately interact with all Nrxns. The HA-Nrxn1::V5-Nrxn3 mouse may serve as a key resource to identify differential protein-protein interactions that underscore the nanoscale architecture observed for each Nrxn. Third, what is the cell-type-specific nanoscale organization of Nrxns at inhibitory synapses? In primary hippocampal cultures, the deletion of *Nrxn3* does not impact inhibitory synaptic transmission, however, when manipulated in vivo and studied ex vivo, a cell-type and synapse-specific role for Nrxn3 becomes apparent[13–15]. Thus, it will be critical to determine the synapse-specific nanoscale organization of Nrxn3 at inhibitory synapses in preparations where the cytoarchitecture is preserved. Our findings provide important insight into how the "molecular code" of individual Nrxns is established and raise critical questions about the underlying mechanisms that control the nanoscale properties of Nrxn1 and Nrxn3.

## Methods

### Mouse generation and husbandry

All experiments utilizing mice followed National Institutes of Health Guidelines and were approved by the Institutional Animal Care and Use Committee at University of Colorado Anschutz Medical Campus. All procedures were conducted in accordance with guidelines approved by Administrative Panel on Laboratory Animal Care at University of Colorado, Anschutz School of Medicine, accredited by Association for Assessment and Accreditation of Laboratory Animal Care International (AAALAC; 00235). Mice were housed kept on a 12-h light-dark cycle with food and water *ad libitum*. The following mouse lines were used: *Nrxn3* JAX: 014157, JAX: 021777 (a generous gift from Dr. Südhof) and *V5-Nrxn3* which we generated here. All mice were genotyped in house. Animals of either sex were either used at P0-1 for culture or P40-50 for tissue slice and biochemistry analysis.

The V5-Nrxn3 mouse was generated in collaboration with National Jewish Regional Mouse Genetics Core using a CRISPR-Cas9 approach to insert a V5 epitope tag into Genomic sequence: NC_000078.7, protein: NP_001185516.2. This tag is inserted in a homologous location as the HA-Nrxn1 tag[25] near splice site 5. Briefly, potential guide RNAs

were determined using an aggregate score of Benchling (RRID:SCR_013955) and CRISPOR[61] ratings and guide RNAs acquired from Synthego were then tested in vitro for cutting efficiency. Micro-injections of zygotes were then completed with the selected guide RNA 5′ GTCTGATTCCTGGCTCCGTG 3′, the homologous directed repair template 5′ GACAGCACCAAACTGAAGAGCCCACTAATTACTTC CCCCATGTTCCGTAATGTGCCCACAGCAGGTAAGCCTATCCCTAACC CTCTCCTCGGTCTCGATTCTACGAATCCCACGGAGCCAGGAATCAGA CGGGTTCCGGGGGCCTCAGAGGTGATCCGGGAGTCCAGCAG 3′, and a Cas9 (Alt-R® S.p. HiFi Cas9 Nuclease V3 IDT Cat# 1081060). Zygotes were then injected into pseudo-pregnant female mice and the result-ing F0 pups were genotyped in house using primers Forward: 5′ CATGTTCCGTAATGTGCCCACACAG 3′; Reverse: 5′ CTGTTTCTTATGGC CGCTCTTGG 3′ (wild type: 295 bps, V5 KI: 337 bps). Germline trans-mission was then established and the mice were bred to homozygosity.

## Plasmids and virus generation

Lentivirus encoding GFP-ΔCre and GFP-Cre have been previously validated and described[8]. shRNA directed against endogenous mouse LRRTM2 was generated with target sequences previously described[5,45] using the following primers Forward: 5′ TTGCTATTCTACTGCGACT CTTCAAGAGAGAGTCGCAGTAGAATAGCATTTTTTC 3′ and Reverse: 5′ TCGAGAAAAAATGCTATTCTACTGCGACTCTCTCTTGAAGAGTCGCA GTAGAATAGCAA 3′ and inserted downstream of a human U6 pro-moter between HpaI/XhoI in a lentivirus transfer plasmid also har-boring a human synapsin promoter driving mCherry. Lentiviruses were produced in HEK293 cells following calcium phosphate transfection with the lentivirus transfer plasmid, pMDL gag/pol, pRSV Rev, and pCMV VSV-G. 24 h after transfection, transfection efficiency was determined by cell fluorescence. Cells were washed 2× with PBS and neuron growth medium was added to the cells. After 24 h the lentivirus containing growth medium was centrifuged for 5 min at 1500xg to pellet cell debris and lentivirus supernatant was either used immedi-ately or aliquoted and frozen at −80 C.

## Cell culture

Primary mouse hippocampal cultures were made from hippocampi isolated from postnatal day 0 or 1 (P0-P1) pups independent of sex as previously described[8]. Briefly, P0-P1 mice were deeply anesthetized on ice before rapid decapitation. Hippocampi were isolated from the brain in Hank's Balanced Salt Solution (HBSS; Sigma Aldrich Cat# H2387), digested in 10U/ml papain (Worthington Biochemical Cat# LS003126) in HBSS at 37 C for 20 min, washed with HBSS and triturated in plating media (MEM) with 10% FBS (Sigma Aldrich), 0.5% glucose (Gibco), 2 mM L-glutamine (GeminiBio Cat# 400-106-100), 0.02% NaHCO₃ (Sigma Aldrich Cat# S5761) and 0.1 mg/ml transferrin (Gemi-niBio Cat# 800-130P-100). Dissociated neurons were plated onto sonicated and acid stripped #1.5 coverslips (Warner Instruments Cat# 64-0714) coated with Matrigel® Basement Membrane Matrix (Corning Cat# 356237) at a density of 200,000 cells per well of a 24 well plate or 400,000 cells per well of a 12 well plate in plating media. Samples for STORM were plated with 100 nm TetraSpeck beads (1:2000; Ther-moFisher Cat# T7279). The day after plating, 70% of the plating media was exchanged with neuronal growth media (MEM) with 2 mM L-glu-tamine, 2.38 mM NaHCO₃, 0.1 mg/ml Transferrin, 0.4% Glucose, 5% FBS and 1:50 Gem21 (GeminiBio Cat# 400-160-010). On day in vitro (DIV) 3–4, 70% of the growth media was replaced with growth media sup-plemented with 4 μM Cytosine β-D-arabinofuranoside (AraC; Sigma Cat# C1768) to arrest glial cell growth. Lentivirus transduction was performed on DIV 3–4.

## Immunocytochemisty

Immunocytochemistry was performed at DIV13-15. For diffraction limited light microscopy, primary neurons from V5-Nrxn3 mice were gently washed with pre-warmed 0.5 M PBS with 1 mM MgCl₂ and

0.5 mM CaCl₂ (PBS⁺/⁺) prior to a 10-min incubation in non-permeabilizing NDS blocking buffer (PBS⁺/⁺ with 2% normal donkey serum) with mouse anti-V5 antibody (1:250, ThermoFisher Cat# 46-0705). Coverslips were then gently washed with ice cold PBS⁺/⁺ three times prior to fixation at RT with 4% PFA and 4% sucrose in PBS for 15 min. The coverslips were then washed 3 times with PBS and per-meabilized with 3 washes of PBS with 0.2% Triton X-100 (PBST), and incubated for 1 hr in blocking buffer (PBST with 2% normal donkey serum). A primary incubation of Guinea pig anti-VGluT1 (Sysy Cat# 135 304) diluted 1:1000 in blocking buffer for 1 h at RT was then completed followed by 3 washes in PBS. Donkey anti-mouse and donkey anti-guinea pig antibodies were diluted 1:1000 in blocking buffer and incubated with the sample for 1 h at RT. Excess secondary was then washed off with three washes in PBS followed by mounting on a cov-erslip with Fluoromount-G (Southern Biotech Cat# 0100-01) for sub-sequent imaging.

All antibodies for STORM imaging, except the chicken anti-Homer1 and guinea pig anti-GluD1[895-932], were directly conjugated to fluorophores. For antibodies not available in directly conjugated pre-parations, the antibody was conjugated using Biotium Mix'n'Stain kits (CF568 Biotium Cat# 92255 and CF647 Biotium Cat# 92449) according to the manufacturer's instructions. We used a previously published live surface labeling protocol to immunolabel surface receptors and adhesion molecules[25]. Briefly, cells were washed gently with HEPES ACSF (in mM: 140 NaCl, 5 KCl, 2 CaCl₂, 2 MgCl₂, 10 HEPES, 10 Glucose adjusted to pH 7.4) followed by a live-surface primary incubation with primary antibodies diluted in HEPES ACSF supplemented with 2% normal goat serum for 20 min at RT. Surface primary antibodies were diluted as follows: mouse anti-GluA1 RH95 clone (Sigma Cat# ZMS1007; conjugated with kit; RH95 clone validated by relative expression by ThermoFisher Cat# MA5-18117) 1:25, mouse anti-V5 (ThermoFisher Cat# 46-0705; conjugated with kit; validated in this paper) 1:25, mouse anti-HA AF647 (ThermoFisher Cat# 26183-A647; validated by knockout[25]) 1:100, and rabbit anti-LRRTM2 AF647 (Bioss Cat# bs-11877R-A647; validated in this paper). Coverslips were then gently washed with HEPES ACSF three times and fixed in 4% PFA 4% sucrose for 15 min at RT. PFA was then quenched for 20 min with PFA Quenching Buffer (PBS with 100 mM ammonium chloride). The cov-erslips were then washed three times with PBS and cells were per-meabilized with three washes of PBST. Nonspecific epitopes were then blocked with STORM Blocking Buffer (PBS with 2% normal goat serum, 0.2% Triton-X100, and 100 mM Glycine) for 1 h at RT. Primary anti-bodies that detected intracellular protein sequences were then diluted in NGS blocking buffer incubated with the sample overnight at 4 C. Primary antibody dilutions were as follows: mouse anti-PSD-95 (Ther-moFisher Cat# MA1-045; conjugated with kit; validated by relative expression by ThermoFisher) 1:50, rabbit anti-RIM (Sysy Cat# 140 003; conjugated with kit; validated by knockout[60]) 1:25, rabbit anti-GluD1[501-600] AF647 (Bioss Cat# bs-12095R-A647; validated by flow cytometry), guinea pig anti-GluD1[895-932] (Frontier Institute Cat# GluD1C-GP-Af840; indirectly stained; validated by knockout[47]) 1:1000, and chicken anti-Homer1 (Sysy Cat# 160 006; indirectly stained; specific for Homer1[62]) 1:1000. Coverslips were then washed three times with PBS followed by a 10-min post-fixation with 4% PFA and 4% sucrose. Cells were washed three times with PBS and then stored at 4 C until the time of imaging.

For immunostaining of only intracellular targets, neurons were first fixed with 4% PFA and 4% sucrose followed by cell permeabilized staining. For indirect staining, the procedure was identical until post-primary incubation washes. After the overnight primary incubation at 4 C, cells were washed three times with NGS Wash Buffer prior to secondary antibody incubation. Secondary antibodies were diluted as indicated: for donkey anti-guinea pig CF568 (Biotium Cat# 20377) 1:500, goat anti-chicken CF568 (Biotium Cat# 20104) 1:500, and goat anti-chicken AF488 (Jackson ImmunoResearch Cat# 103-545-155) 1:500. Coverslips were then washed three times with NGS Wash Buffer

(PBST with 0.2% NGS), PBS once and then post-fixed for 10 min with 4% PFA 4% sucrose in PBS. Three PBS washes were then completed, and the coverslips were stored until the day of imaging.

## STORM imaging

3D dSTORM bypasses traditional light diffraction limitations by permitting precise localization of single molecules. The Double Helix Optics SPINDLE® and phase masks generate a double-helix point spread function, which provides a 15 nm xy- and 30 nm z-resolution in addition to a 3 μm z-range[18]. 3DTRAX® software (Fig. S1a) and previously published code[19], enabled the quantitative assessment of single molecule localizations of synaptic proteins and nonuniform high-density SSDs (Fig. 1a). STORM imaging was completed on a Nikon N-STORM system at the BioFrontiers Institute Advanced Light Microscopy Core (RRID: SCR_018302) equipped with a 100x TIRF objective (NA1.5), Agilent Technologies MCL400B (405, 488, 561 and 647 nm laser lines), and a Hamamatsu ORCA-Flash4.0 V2 attached to a Double Helix Optics (Boulder, CO) SPINDLE®. Typical incident laser power measured out of the objective was ~30 mW for the 647 channel and ~25 mW for the 561 channel. Double Helix phase masks DH1-670-2045 and DH1-580-2045 were used to generate Double Helix point spread functions (PSF) for high precision z localization of PSFs for the 647 nm and 561 nm laser lines respectively. Emission filters used were Chroma ET670/50 m for the 647 laser and ET600/50 m for the 561 nm laser line.

Calibration stacks of Double Helix PSFs were collected from 100 nm TetraSpeck microspheres deposited on a coverslip. The individual phase masks were aligned using a Bertrand Lens after which the PSFs were manually inspected to ensure equal intensity of Double Helix PSF lobes. Then, a region of interest containing as many non-overlapping Double Helix PSFs as possible was selected and a calibration stack with a range of 4μm at 50 nm steps and 10 acquisitions per step were acquired. Calibrations for each experiment were completed in Double Helix 3DTRAX software to allow for 3D localization of experimental PSFs. These calibration stacks were also used to generate the two-channel alignment correction for each experiment.

After calibration, coverslips were placed in a Warner Scientific Instruments Quick Release imaging chamber (Model QR-41LP) and immersed in 1 mL of photo-switching buffer[63,64]. An optimal region of interest, containing secondary and tertiary dendrites of neurons with pyramidal morphology was selected and a widefield image was taken prior to STORM imaging. For STORM imaging, the lasers were positioned in a highly inclined angle (HiLo) near but below the angle at which TIRF was observed, and 20,000 frames were collected with Nikon Perfect Focus on and an exposure time of 30 ms.

All Double Helix PSF localization, fiducial marker drift correction, and two channel alignment was completed using Double Helix Optics 3DTRAX® Software. In cases where emission lasted longer than one frame only the first frame containing the emitter was included for further analysis. Emitters with precision values greater than 40 nm in the XY dimensions or 80 nm in the Z dimension were excluded from further analysis.

## STORM analysis

The experimenters were blinded for all steps of analysis until data was collated for statistical analysis. Synapses were identified using a custom R script utilizing a non-biased DBScan method with an epsilon of 100 nm and minimum points of 10 as has been widely been used in the past[5,19,25,63]. All synapses were then validated manually to ensure that each potential synapse cluster contained only one synapse, that the localizations appeared synaptic (roughly a disk shape), and that the cluster was not a Tetraspeck bead. Identified synapses were then re-clustered using a mean minimal distance method[19] cutoff of 4 to remove localizations or additional synapses that were far from the synapse center using a custom R script. This reclustered synapse data was then subjected to final analysis using previously published MATLAB scripts[19]. For GluA1, PSD-95, RIM1, and GRID1 nanocluster_detection_3D thresholds of T = 2.5, cutoff = 100 nm, and density standard deviation cutoff of 4 was used. Synaptic compartments were excluded from further analysis if they contained no SSDs for GluA1, PSD-95, or RIM1. It was found that V5-Nrxn3, HA-Nrxn1, and LRRTM2 formed discrete high-density areas that interfered with SSD detection using these parameters and instead cutoffs of T = 2, cutoff = 100 nm, and density standard deviation cutoff of 1.5 was used. The enrichment index for protein enrichment was calculated as (Σprotein enrichment 20–60 nm)/(the number of observed radii, 3) and enrichment index for cross-correlation was calculated as (Σcrosscorrelation(10 nm–50 nm))/(9, the number of observed radii, 9). Synapses with more than 5 localizations and at least 1 SSD for V5-Nrxn3, HA-Nrxn1, LRRTM2, and GRID1 were considered positive for that protein. For transsyanptic alignment, SSDs that were incorrectly translated were excluded from further analysis. In all experiments, a manual count of SSDs seen in synapses was compared to the calculated values to ensure the fidelity of detection.

The alphavol function[19] was used to determine the volume of the synapse and SSDs. For compartment volume, the input to alphavol was all the synaptic localizations for each protein. To calculate average SSD volume, the alphavol of each SSD in a synapse was calculated and then averaged for that synapse. To determine the relative density of synaptic proteins in SSDs, the density of localizations in SSDs was divided by the density of localizations in the entire synaptic compartment.

For SSD nearest neighbor analysis, the SSD centers were first extracted from the nanocluster_detection_3D analysis. For experimental conditions, the MATLAB knnsearch function was used on the set of SSDs in both channels. To generate a random selection of SSD centers, we employed previously published scripts[19]. Briefly, the localizations of one channel were randomized to an even distribution within the compartment volume using the get_cluster_randomized functionand then random SSD centers were picked from this distribution[19]. For each synapse, the number of selected randomized SSDs was equal to the number of experimental SSDs.

To determine the radial distribution of localizations SSDs at synapses, the pdist2 function was used as has been used in the past[25]. Briefly for comparisons of presynaptic proteins to Homer1 centroid, either localization coordinates or SSD centers were assayed for their distance to the centroid (mean point) of Homer1.

For quantification of Nrxn SSD overlap, two complementary methods were utilized. The first method was to quantity the overlap the masks generated by SSDs in each channel. This method is more likely to overestimate the amount of overlap because the polygon mask of the SSD points may include volume where no localizations are present. Briefly, the MATLAB function delaunayTriangulation was used to generate polygons of each SSD and a mask was generated of SSDs for Nrxn1 and Nrxn3 by generating a mesh with voxel size of 10 nm and testing whether each voxel of the mesh contained a portion of the SSD. The masks for Nrxn1 and Nrxn3 were then multiplied to obtain a mask of only the overlapping points. The volume of the overlapping mask was determined by summing the number of overlap mask voxels and the volume of each channel was calculated as the sum of their individual masks. For the second method, the SSD overlap was determined as the volume of SSD points in channel 1 that reside within the SSD volume of channel 2. Briefly, alphaShape was used to determine the SSD boundaries of channel 2 and then each SSD localization of channel 1 was tested for inclusion inside a channel 2 SSD. The alphaVolume of overlapping localizations from channel1 was then determined for each SSD and the sum of SSD overlap for every SSD in channel 1 value was divided by the total of SSD volumes for channel 1 in that synapse.

## Simulations

Quantal release of glutamate was carried out with MATLAB (Version R2021a; MathWorks, Natick, MA, USA) by using a Monte Carlo algorithm that simulated the stochastic behavior of molecule diffusion and dynamic binding to AMPARs in a complex microenvironment with a time step of 0.5 μs.

## Simulation of AMPAR distribution

Considering its unique structure, the CA1 could be considered as a large parallel arrangement of a few hundred active zones aligned to the corresponding postsynaptic sites[36]. Thus, the extracellular space between the presynaptic and postsynaptic membrane was regarded as two paralleled coaxial cylinders of 0.5 μm length, with a 15 nm synaptic cleft between the cylinders[31].

We adopted a previous model to describe the number of different internal states of GluA1[37]. The radiuses of nano- and synaptic clusters of GluA1 were calculated based on STORM data. A total of 100 GluA1 were placed as assigned with the ratio estimated from STORM (Table 1).

## Simulation of glutamate release and AMPAR activation

The initial fusion pore conductance of a single vesicle is >375 pS[33] and the relationship between transmitter flux and conductance[38] permits a calculation of vesicular expansion time ($\tau = 73\gamma^{-1}$ μs, where γ is the fusion pore conductance in nS) of 0.2 ms to release its all transmitters. The number of glutamate molecules in the vesicle was set to 3000. After release, glutamate molecules do Brownian motion at a diffusion rate of 0.4 μm$^2$ / ms.

When glutamate hits the membrane or even AMPAR, it will not be bound but reflected. A 9-states AMPAR reaction scheme was taken from a previous study[37]. An AMPAR was run against the glutamate transients to calculate the open probability of individual AMPAR. Every AMPAR was regarded as a circular area with a radius of 10 nm and its internal state depending on the number of glutamates hitting this area during a one-time step. The effect of glutamate binding to GluAs is negligible for the much greater number of glutamate molecules (3000) than GluA1 (100). The rate constants of GluA1 were initially set as previous study[37] and were adjusted within the constraint of microscopic reversibility[39]. Transporters were uniformly distributed on the glial sheath which surrounded the synapse. The default density of transporters was 2000 / μm$^2$ and the distance between synaptic edge and glia was 40 nm[31].

The traces shown here were mean values of 160 runs with release sites randomly distributed through the active zone with a radius equal to nanocluster. All the default parameters we used are listed in Table 1 unless otherwise stated. The 10–90% rise time and decay time were calculated by fitting the rise and decay phases of EPSCs with double exponential functions. EPSC at time t is generated by

$$I(t) = [g \times n(t)] \times (V_m - V_{GluAs}) \qquad (1)$$

where g is the single-channel conductance set at 31 pS for GluA1[35], $n(t)$ is the number of open GluA1 at time $t$, $V_m$ is the resting membrane potential and $V_{GluAs}$ is the reversal potential of GluA1.

## Immunoprecipitation

Male and female mice were deeply anesthetized with isofluorane vapor before rapid decapitation. Whole brains were quickly transferred to a solution of 1% Triton X-100 buffer with protease inhibitors and homogenized via a mortar and pestle. The homogenate was then spun at 90,000 g for 30 min at 4 C to remove debris. The resulting supernatant was then collected and pre-cleared with 0.1% Triton X-100 washed protein-A beads (RepliGen Cat # 10-1003-01) for 1 h at 4 C. A small portion of pre-cleared supernatant was saved for input, and the remaining supernatant was then incubated with 2 μg of Rabbit anti-V5-tag antibody (Cell Signaling Technology Cat# 13202) or 2 μg of control rabbit IgG (Jackson ImmunoResearch Cat# 309-005-008) and 30 μL of protein-A beads for 4 h rotating head over tail at 4 C. The antibody bound beads were then washed three times with 0.1% Triton X-100 and pelleted to isolate the protein bound beads. The supernatant was then removed and the bead bound proteins were eluted with 100 μL of 1x sample buffer and boiled for 5 min at 100 C.

## Hippocampal lysate collection

Male and female mice were deeply anesthetized with isofluorane vapor before decapitation. Hippocampi were isolated from P40-50 V5-Nrxn3$^{+/+}$ and wildtype littermates independent of sex and homogenized via sonication in RIPA buffer with protease inhibitors. Protein levels were then normalized via a BCA Assay (ThermoFisher Cat# 23227) and stored at −80C until further processing.

## Immunoblotting

Proteins were resolved by SDS-PAGE with 7.5%–10% polyacrylamide gels. Proteins were then transferred onto nitrocellulose membranes at 300 mA for 2 h at 4 C. Verification of transfer was performed with a Ponceau stain that was subsequently washed off before blocking for 1 h at RT with 5% milk in PBS. Primary incubations were diluted into PBS with 5% milk and 0.1% tween and incubated overnight at 4 C. Primary antibodies were diluted as follows: rabbit anti-panNrxn (Frontier Institute Cat# MSFR104630) 1:1,000, mouse anti-V5 (ThermoFisher Cat# 46-0705) 1:1,000, guinea pig anti-RIM1/2 (Sysy Cat# 140 205) 1:1,000, mouse anti-PSD-95 (ThermoFisher Cat# MA1-046) 1:2,000, rabbit anti-Homer1 (Sysy Cat# 160 003) 1:1,000, mouse anti-GluA1 (Sigma Cat# ZMS1007) 1:1,000, mouse anti-GluA2 (BioLegend Cat# 810501) 1:1000, and mouse anti-βActin (Millipore Sigma Cat# 810501) 1:10,000. Membranes were then washed three times with PBS prior to incubation with infrared AF680 or AF790 secondary antibodies from Jackson ImmunoResearch diluted 1:25,000 in blocking buffer for 1 h at RT. Excess secondary was washed off with three PBS washes prior to imaging on a LI-COR Odyssey FC system. Analysis was performed using ImageStudio (LI-COR).

## Whole cell patch clamp electrophysiology

P40-50 animals were deeply anesthetized using isoflurane and decapitated. The mouse brain was then quickly dissected and 300 μm horizontal slices were collected using a Leica VT1200 vibratome in an ice cold high-sucrose solution that consisted of 85 mM NaCl, 75 mM sucrose, 25 mM D-glucose, 25 mM NaHCO$_3$, 4 mM MgCl$_2$, 2.5 mM KCl, 1.3 mM NaH$_2$PO$_4$, and 0.5 mM CaCl$_2$. Slices were then transferred oxygenated ACSF containing 126 mM NaCl, 26.2 mM NaHCO$_3$, 11 mM D-Glucose, 2.5 mM KCl, 2.5 mM CaCl$_2$, 1.3 mM MgSO$_4$-7H$_2$O, and 1 mM NaH$_2$PO$_4$ at 31.5 C for 30 min followed by recovery at RT for 1 h before recordings. Slices were superfused with 29.5 C ACSF containing 0.5 μM tetrodotoxin and 100 μM picrotoxin to isolate mEPSCs. Internal solution consisted of 115 mM Cs-methanosulfonate, 15 mM CsCl, 8 mM NaCl, 0.2 mM EGTA, 10 mM HEPES, 4 mM Mg-ATP, 0.3 mM Na-GTP, 10 mM TEA-Cl, 10 mM Na2-phosphocreatine, and 1 mM MgCl2 which resulted in an osmolality of ~290 mOsm[9]. Cells were held at −70 mV for recordings and the resulting traces were analyzed in Clampfit 10 (Molecular Devices). Miniature events were identified using template matching and confirmed by experimenters.

Culture electrophysiology recordings were performed under the same conditions. Whole-cell voltage-clamp recordings were performed on pyramidal neurons in dissociated hippocampal culture, which were identified by the presence of dendritic spines.

## qRT-PCR of *Nrxn3* mRNA in dissociated cortical cultures

For validation of *Nrxn3* cKO, qRT-PCR with a β-actin internal control was performed. mRNA from *Nrxn3* cKO cultures infected with either

inactive GFP-ΔCre or GFP-Cre were harvested at DIV13-15 with a Quick-RNA Micro-Prep mRNA isolation kit (Zymo Research). The concentration of mRNA was then normalized between samples and RT was completed using qScript XLT 1 Step RT-qPCR ToughMix Kit (Quanta-bio) on a Bio-Rad CFX384 Real-Time System in a TempPlate 384-Well Full Skirt PCR plate (USA Scientific). The qPCR primers and probes used were: β-actin: F: GACTCATCGTACTCCTGCTTG, R: GATT ACTGCTCTGGCTCCTAG, Probe: CTGGCCTCACTGTCCACCTTCC. Nrxn3β: F: ACCACTCTGTGCCTATTTCTATC, R: TGTGCTGGGTCT GTCATTTG, Probe: TCGCTCCCCTGTTTCCCTTCG.

## Statistics and reproducibility

All statistical analysis was complete in GraphPad Prism 9. Assumptions of tests, such as normality and homoscedasticity were evaluated before the application of the tests. If outlier tests were applied, the same outlier test with the same threshold was used for all data of both groups. All data contain at least 3 biologic replicates (N = independent animal or culture) except where otherwise noted.

## Reporting summary

Further information on research design is available in the Nature Portfolio Reporting Summary linked to this article.

## Data availability

Source data generated in this study are provided in the Source Data file. Raw data will be provided upon request.

## Code availability

The custom MATLAB script used for the computational modeling of the impact of *Nrxn3* on AMPAR EPSCs can be downloaded from Github: https://github.com/Han-y/Synapse-Model-for-Aoto-Lab. The custom scripts for STORM analysis can be downloaded from Github: https://github.com/Brian3342/Aoto_Lloyd_2023.

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

## Acknowledgements

We would like to thank the Aoto Lab for helpful discussions and advice. We thank Dr. Kevin Crosby, Dr. Samantha Olah, Mr. Michael Johnson, and Dr. Jian Wei Tay for aid in computational analysis and coding. We thank Dr. Warren Colomb, Dr. Joseph Dragavon, and Mr. Scott Gaumer for technical and software support. We thank Dr. Jennifer Matsuda and the National Jewish Regional Mouse Genetics Core Facility for aid in the generation of the V5-Nrxn3 mouse. We thank Dr. Thomas Südhof for the generous gift of the 2xHANrxn1 mouse. Super-resolution microscopy was performed at the Bio-Frontiers Institute Advanced Light Microscopy Core (RRID: SCR_018302) on a Nikon N-STORM microscope supported by the Howard Hughes Medical Institute. This work was supported by the following grants from Natural Science Foundation of China 82022018, 32070958, 82161138025, to B.Z.; Shenzhen-Hong Kong Institute of Brain Science-Shenzhen Fundamental Research Institutions 2023SHIBS0004 to B.Z., Guangdong Pearl River Funding to B.Z., and the NIH 3T32GM007635-41S1 to B.A.L., R01MH116901 to J.A. and R21MH129620 to J.A.

## Author contributions

B.A.L. prepared primary hippocampal cultures and performed immunostaining, protein biochemistry, 3D dSTORM and data analysis. R.R. performed and analyzed ex vivo slice electrophysiology, performed Western blot and immunocytochemistry for confocal microscopy. Y.H. performed synapse modeling in the lab of B.Z. J.A. performed culture electrophysiology. B.A.L. and J.A. are responsible for study conception, experimental design, and data interpretation. B.A.L. and J.A. wrote the manuscript with input from all authors.

## Competing interests

The authors declare no competing interests.
