## [Peer Review File · Nature Communications]

REVIEWER COMMENTS

Reviewer #1 (Remarks to the Author):

Lloyd et al systematically characterized the nanoscale organizations of Nr3 and Nr1 together with a few other key synaptic proteins and demonstrated an interesting model that they preferentially align with different postsynaptic receptors to form distinct nanocolumns. This is the first study that identifies the role of a presynaptic adhesion molecule in modulation of synaptic nanoarchitectures. The results are very informative. I have only a few comments to help improve it.

Major:

1. In figure1, despite that the nanoscale organizations are essential for synaptic transmission, the synaptic retention of AMPARs and overall receptor density within the PSD compartment (especially the later) are major determinants for synaptic strength. The authors should carefully quantify these, compare them with previous confocal data, and incorporate them in the model simulation.

2. The authors tried to suggest that Nr3 couples to LRRTM2 and its deletion leads to similar changes in synaptic structure and functions. However, Ramsey 2021 paper showed that changes in nanoorganization of receptors alone by LRRTM2 disruption impact only evoked but not the miniature EPSCs, while the previous work of the senior author in 2015 showed that Nr3 KO reduces both mini and evoked currents. The authors should discuss the potential reasons. Is it possible that a change in overall receptor number or density accounts for the different functional impacts?

3. The description of the model parameters is confusing.

1) In line 137, what parameter did the authors adjust to represent the 17.4% decrease in the number of GluA1 SSDs? Are there multiple SSDs within one synapse in the model? If so, how were they distributed within the PSD?

2) The statement in lines 140-141 makes no sense. To my understanding, the 33% decrease in GluA1 enrichment opposite RIM1 SSD is the net output of two factors: a) the 19.2% decrease in GluA1 density within SSDs and b) an unmeasured lateral displacement between SSDs of RIM and GluA1. Therefore, the 33% decrease in GluA1 enrichment opposite RIM1 SSD should be the key factor to be tested in the model. The 19.2% decrease in GluA1 density within SSDs is less relevant.

3) It's unclear what parameters exactly was changed in the third test.

4. Key information is missing for the quantification of alignment between PSD-95 and V5-Nr3 or HA-Nr1. The alignment analysis is based on the assumption that the general shapes of AZ and PSD are very similar and parallel with each other. Since Nr3s distribute in form of tight small clusters and could not form the stereotyped disc shape of AZ as RIM1 does, the relative positioning of SSDs in Fig 4g and 5g would strongly depend on the arbitrary view angle. Also, this makes the transformation and the following enrichment analysis unreliable. The authors need to provide more details on how to avoid this.

5. The authors found that more than 90% of excitatory synapses are Nr1 positive and there are averagely 2 Nr1 SSDs per synapse. These numbers are quite different from previous study using the same KI mice (Trotter 2019) showing that only 40% synapses have Nr1 and most Nr1+ synapses have only one Nr1 nanocluster. How the authors interpret these differences?

6. How is the enrichment index calculated?

Minors:

1. In abstract line31, Nr3 could be the first presynaptic adhesion molecule identified to be required for nanocolumns but certainly not the first presynaptic molecule.

2. The order of citations were messed up. Some references, such as ref17, were never cited in the text.

3. Line 191, should be "the mean radial distance of V5-Nrxn3..." In the same sentence, the numbers represented in Fig 4C is missing.
4. I'm troubled by the overused "enrichment". For example, in line 250-251, there are two enrichments, one for a protein and the other for SSDs, both different from the context of "enrichment analysis/index".

Reviewer #2 (Remarks to the Author):

While we know much about the molecular machinery within each compartment, we are only beginning to understand how these compartments are structurally in register and functionally integrated with one another. It is likely that this architecture may allow for precise synaptic information exchange and may be modulated to contribute to the remarkable plasticity of brain function. Therefore this manuscript is timely and important. The main findings of this work, that NRXN1 and 3 are organized in specialized and distinct clusters, which allow them to retain their individual properties and function in parallel, are important to understand how synapses assemble and work.

Overall, this is an extremely data-rich manuscript with experiments carefully planned and executed. However, it is a bit hard to read, perhaps because most of the time the authors don't elaborate on what the measurements mean. For example in figure 1A-E, the authors describe variation in synaptic volume occupied by GluA1, number of GluA1 SSDs, GluA1 SSD volume, density, etc. Some parameters don't change while others do. But there is no explanation of what this all means, even if just structurally. Another example is at lines 152-157: Nrxn3 KO significantly alters the PSD-95 synaptic parameters. What conclusion can they draw from this that can drive the next set of experiments? More information to guide the reader on what each parameter/experiment is telling is crucial to follow the progression of experiments, otherwise the manuscript looks disjointed and a little descriptive.

Introduction is well crafted and contains all the necessary bits to follow the experiments laid in the result section.

Line 31 in the abstract (and line 70 and 125): what do the authors mean when they write that NRXN-1 is "the first presynaptic molecule required for excitatory synapse nano-organization"? That NRXN-1 is the first to be expressed, or that is "the first being identified" for the nano-organization? Please clarify.

I presume NXN3 KO deletes both alpha and beta NRXN, but this is not stated, please specify.

Perhaps the modelling of synaptic transmission (Figure 1) could be strengthened by actual ephys experiments using at least some of the different conditions utilized so that all of these SSDs findings can have a physiological interpretation.

Figure 1L: Ablation of Nrxn3 reduced the density of GluA1 protein opposite RIM1 SSDs by more than 33%. Figure L shows a reduction from ~ 1.8 down to ~ 1.6 , which is only $\sim 12\%$. Or does this refer to figure 1D? Please explain the discrepancy

Figure 2 e-g: what's the difference between these graphs and the one shown in Figure 1 f-I?

I have no comments on the generation and testing of the V5 tagged NRXN mouse line, other than praising the authors for the effort in generating a new mouse line that will help understanding NRXN expression, function and physiology

Please explain why in figure 4, V5-Nrxn3 was first co-stained with Homer1 instead of PSD95, as they both identify excitatory synapses.

Figures need to be numbered on the file, otherwise it's too difficult to follow which experiments is referred to in the text.

Line 278: please explain what do you mean by "non-redundant transsynaptic signaling."

Finally, an overall schematic model of the findings would be helpful to wrap it all up.

Reviewer #3 (Remarks to the Author):

Neurexins (Nrxns) are presynaptic adhesion molecules that bind to multiple postsynaptic partners to control the specification of various types of central synapses. Nrxns have a large number of isoforms that distinctly interact with different postsynaptic binding partners. Importantly, the repertoire of presynaptic Nrxn isoforms and their postsynaptic binding partners at individual synapses differ synapse by synapse, depending on cell types, neuronal circuits, and brain regions at tissue levels. However, how this diverse repertoire specifies the function of individual synapses largely remains unknown. The authors address this important question by nano-scale molecular imaging of hippocampal primary cultures with super-resolution microscopy. The authors demonstrated that Nrxn3, but not Nrxn1, was required for trans-synaptic organizations of RIM1 and AMPAR nanoclusters at excitatory synapses, which can play an important role in glutamatergic transmission. By generating a new mouse line expressing both HA-tagged Nrxn1 and V5-tagged Nrxn3, the authors also revealed that presynaptic nanoclusters of Nrxn1 and Nrxn3 were segregated from each other, and distinctly arranged close proximity to postsynaptic nanoclusters of GluD1 and LRRTM2, respectively. Their experimental technique is state of art, and nicely visualizes subsynaptic nano-scale clusters of synaptic proteins on the presynaptic and postsynaptic membrane. However, this study seems to be descriptive because it lacks a mechanistic explanation to the segregated distribution of Nrxn1 and Nrxn3 nanoclusters on the presynaptic membrane. For instance, does this segregation depends on the difference between Nrxn1 and Nrxn3 isoforms themselves or dominant recruitments of Nrxn1+SS4(interacts with Cbln-GluD1) and Nrxn3-SS4(interacts with LRRTM2) into Nrxn1 and Nrxn3 nanoclusters, respectively? It also remains unclear whether Nrxn1 and Nrxn3 nanoclusters actually interact with GluD1 and LRRTM2 nanoclusters in a segregated fashion. In addition, the authors should address the following specific points.

#1 The authors' conclusion is limited in scope because this study tested hippocampal culture synapses only. To make the current conclusion more generalized, the authors should test different types of synapses with distinct repertoires of Nrxns in other neuronal cultures or brain tissues. In this respect, the statement "Nrxn3 is the first functionally-relevant presynaptic adhesion molecule responsible for organizing the nanoscale architecture of excitatory synapses" sounds too strong to me.

#2 GluD1 may have not only the cell-surface pool but also the intracellular reserve one, as in the case of GluAs. However, the current data cannot distinguish the former from the latter at individual synapses, making it difficult to judge signals presented in this study as the cell-surface GluD1 nanoclusters. The authors should test the cell-surface GluD1 by surface labeling with antibodies against its extracellular domain or demonstrate the current signals are selectively associated with the cell-surface pool, but not the reserve pool.

#3 The validation of anti-LRRTM2 antibody is not convincing. The current data cannot rule out a possibility that the remaining labeled puncta in LRRTM2-knockdown neurons are non-specific signals. The authors should validate this antibody with more appropriate controls such as LRRTM2-KO mice.

#4 Although this study is highly dependent on immunocytochemistry, the current manuscript also

lacks the information about the specificity of immunocytochemical signals. The authors can present more information about the signal specificity under the authors' experimental conditions.

Response to Reviewers

Reviewer #1 (Remarks to the Author):

Lloyd et al systematically characterized the nanoscale organizations of Nrx3 and Nrx1 together with a few other key synaptic proteins and demonstrated an interesting model that they preferentially align with different postsynaptic receptors to form distinct nanocolumns. This is the first study that identifies the role of a presynaptic adhesion molecule in modulation of synaptic nanoarchitectures. The results are very informative. I have only a few comments to help improve it.

We appreciate the reviewer's positive interpretation of our work.

Major:

1. In figure 1, despite that the nanoscale organizations are essential for synaptic transmission, the synaptic retention of AMPARs and overall receptor density within the PSD compartment (especially the later) are major determinants for synaptic strength. The authors should carefully quantify these, compare them with previous confocal data, and incorporate them in the model simulation.

We thank the reviewer for this comment and have modified our model. We agree that the overall receptor density is important in the PSD compartment. We quantified the absolute AMPA receptor density in PSD compartments and found that the overall density of AMPARs was increased by 29% in the Nrxn3 KO condition. Importantly, this increase in AMPAR density in the PSD compartment was coincident with a significant 32% reduction in compartment volume. Computational modeling indicated that the increased GluA1 compartment density was not likely driven by the stable relocation of AMPARs out of SSDs and into other regions of the compartment (reduced SSD retention) but the result of the 33% decrease in GluA1 density opposite RIM1 SSDs and the decreased compartment volume. Thus, there are actually fewer total GluA1 localizations per PSD compartment volume in Nrxn3 KO synapses relative to controls.

While the precise mechanism that reduces AMPARs in the PSD compartment and in SSDs is difficult to test directly, our current data are congruent with our previous confocal data that indicated that the surface stability of GluA1 and the size of GluA1 puncta are reduced. Removal of AMPARs in the *Nrxn3* KO may be via direct endocytosis or via diffusion of GluA1 out of SSDs (e.g. reduced retention) followed by endocytosis.

In our updated computational model (Figure 3), we segregated the synapse into three regions: 1. The core of the SSD (0-40nm), a peri-SSD region (an annulus with an inner radius of 40 and an outer radius of 80) and a peripheral, non-SSD region (annulus with an inner radius of 80 and an outer radius of 250nm). These values approximate the dimensions of wild-type synapses that we quantified. We then populated AMPARs into each ring proportional to our experimentally determined AMPAR densities for control hippocampal synapses (AMPARs per ring: Ring 1: 4; Ring 2: 6; Ring 3: 3) and simulated the AMPAR EPSC amplitudes in response to release from a RIM1 SSD for each ring while taking into account AMPAR activation kinetics and the rate of glutamate diffusion. Consistent with previous work that suggested that the majority of evoked transmission occurs in synaptic nanocolumn SSDs and that the density of AMPARs is critical at these regions (Nair et al., 2013; Tang et al. 2016; Ramsey et al. 2021), we found that AMPARs in the non-SSD region contribute <10% of the total AMPAR EPSC (Figure 3e). We now include in our model the contribution of non-SSD AMPARs as well as the impact of the condensed PSD compartment in addition to the 33% decrease in GluA1 enrichment and the reduction of the number of SSDs. Together, these parameters result in a ~36% reduction in AMPAR EPSC amplitude.

2. The authors tried to suggest that *Nrx3* couples to LRRTM2 and its deletion leads to similar changes in synaptic structure and functions. However, Ramsey 2021 paper showed that changes in nanoorganization of receptors alone by LRRTM2 disruption impact only evoked but not the miniature EPSCs, while the previous work of the senior author in 2015 showed that *Nrx3* KO reduces both mini and evoked currents. The authors should discuss the potential reasons. Is it possible that a change in overall receptor number or density accounts for the different functional impacts?

This is an excellent point and builds on the elegant work of Ege Kavalali and others who have determined that synchronous, asynchronous and spontaneous release occur in distinct pathways. There are fundamental differences in the experimental approaches that may explain this discrepancy.

1. Here, we use a conditional knockout model that results in undetectable mRNA levels of neuroligin-3 after 14 days post deletion. Ramsey et al., 2021 relied on shRNA to knockdown endogenous LRRTM2. Although the shRNA used is highly efficient (we used the same shRNA sequences here!) shRNA suppresses but does not eliminate endogenous mRNA and protein expression. The remaining endogenously expressed LRRTM2 may be sufficient to sustain mEPSCs.
2. Ramsey et al., 2021 used transient transfection to knockdown endogenous LRRTM2 and replace with a proteolytically sensitive LRRTM2. Although gross morphology (e.g. spine properties and synapse numbers) was unaffected, the exogenous LRRTM2 was overexpressed for at least 7 days – it is estimated that transient transfection introduces roughly 10^3 to 10^5 plasmids per nuclei, which may result in substantial overexpression of an exogenous protein (Cohen et al., 2010). Prolonged overexpression of LRRTM2 prior to acute proteolytic cleavage might have unintended consequences on synaptic transmission.
3. In Ramsey et al., 2021, the nanoscale AMPAR properties were determined by using overexpressed SEP-GluA1 and not endogenous GluA1, which we studied here. N-terminally tagged GluA1 (GFP-GluA1) interferes with the constitutive trafficking of GluA1. GFP-GluA1 did not contribute to basal AMPAR-mediated synaptic transmission and only trafficked into the synapse following the induction of LTP (Díaz-Alonzo et al., 2017). At the nanoscale level, does exogenously expressed SEP-GluA1 fully recapitulate the nanoscale properties of endogenous GluA1?
4. The functional measurements were performed at different periods post-manipulation. We previously deleted *Nrxn3* starting on DIV 4 and monitored evoked synaptic transmission and mEPSCs 10 days later in primary cultures. In Ramsey et al., 2021, evoked and mEPSCs were monitored immediately after the acute proteolytic cleavage of LRRTM2 - up to 15 minutes post cleavage. Nair et al., 2013 demonstrated that AMPAR SSDs exhibit differences in stability – some are stable for tens of minutes

while others disappeared after <5 minutes. One possible explanation is the 10-day deletion of Nrnx3 results in the depletion of endogenous AMPARs from most SSDs as well from the PSD compartment (quantified in Figure 1). By contrast, the acute proteolytic cleavage of LRRTM2 may primarily affect the transient population of AMPAR SSDs, while leaving the overall density of GluA1 in the compartment unaltered.

5. The ablation of Nrnx3 eliminates transsynaptic interactions and Nrnx3-dependent signaling with all postsynaptic ligands, including interactions with LRRTM2 *and* LRRTM1. The loss of Nrnx3-dependent signaling may explain the functional differences. Indeed, Bhoury et al., 2018 observed identical synaptic phenotypes – impaired AMPAR mediated EPSCs and reduced mEPSC amplitudes following in LRRTM1 and LRRTM2 double knockout neurons in hippocampus. Unfortunately, we were unable to find a reliable anti-LRRTM1 antibody for immunocytochemistry extracellular labeling.

6. Unlike the Nrnx3 KO phenotype reported by confocal microscopy, Ramsey et al., did not report a reduction in surface SEP-GluA1 expression following the acute manipulation of LRRTM2. Assuming SEP-GluA1 accurately reflected endogenous AMPARs, this suggests that, consistent with the model proposed in Guzikowski and Kavalali, 2021, spontaneous transmission is not necessarily dependent on GluA1 density in SSDs but dependent on GluA1 density in the PSD compartment.

We have expanded our discussion to include these points.

3. The description of the model parameters is confusing.

We apologize for the confusion. We have expanded our computational modeling to include the contribution of non-SSD AMPARs (new Figure 3) and discuss in detail what each parameter means and how they were modeled.

1) In line 137, what parameter did the authors adjust to represent the 17.4% decrease in the number of GluA1 SSDs? Are there multiple SSDs within one synapse in the model? If so, how were they distributed within the PSD?

Our model uses a synapse containing a single transsynaptic nanocolumn to model the 17.4% decrease in GluA1 SSDs, we ran the simulated values 10x for control synapses and 8.2x for Nrnx3 KO synapses.

2) The statement in lines 140-141 makes no sense. To my understanding, the 33% decrease in GluA1 enrichment opposite RIM1 SSD is the net output of two factors: a) the 19.2% decrease in GluA1 density within SSDs and b) an unmeasured lateral displacement between SSDs of RIM1 and GluA1. Therefore, the 33% decrease in GluA1 enrichment opposite RIM1 SSD should be the key factor to be tested in the model. The 19.2% decrease in GluA1 density within SSDs is less relevant.

We have altered our model to account for the 33% reduction in GluA1 density. As described above, we incorporated the 33% reduction in the annuli between 0-100nm, which comprises the SSD and peri-SSD regions.

3) It's unclear what parameters exactly was changed in the third test.

We have modified the computational model and discussed which parameters are changing in each experiment.

4. Key information is missing for the quantification of alignment between PSD-95 and V5-Nrx3 or HA-Nrx1. The alignment analysis is based on the assumption that the general shapes of AZ and PSD are very similar and parallel with each other. Since Nrnx distribute in form of tight small clusters and could not form the stereotyped disc shape of AZ as RIM1 does, the relative positioning of SSDs in Fig 4g and 5g would strongly depend on the arbitrary view angle. Also, this makes the transformation and the following enrichment analysis unreliable. The authors need to provide more details on how to avoid this.

We share the reviewer's concern about the transsynaptic translation, which is why we also utilized the nearest neighbor analysis in Fig 5h&i and Fig 6h&l, which measured the absolute distances between Nrnx SSDs and PSD-95 SSDs. These measurements did not use any type of translation similar to the

enrichment analysis and showed similar results that Nrnx3 is more tightly coupled to PSD-95 than Nrnx1.

The reviewer raises an important point about the Nrnxns not organizing into a disk shape similar to RIM1. While V5-Nrnx3 and HA-Nrnx1 both form discrete SSDs, there are additional points outside of the SSDs (5g and 6g) from which the translation vector can be calculated. One advantage of 3D-dSTORM is that it allows us to view the data in any orientation and therefore Fig 5g and Fig 6g were rotated such that the synapse is viewed directly en face. Below are example V5-Nrnx3 and HA-Nrnx1 synapses. We present the data in two forms: the first represents the localizations in the acquisition orientation and after 3D rotation to an en face view.

5g Acquisition Orientation

V5-Nrxn3 SSDs

PSD-95

Merge

PSD-95 Normalized Density
 ≤50 100 ≥150

100nm

5g En Face

100nm

6g Acquisition Orientation

HA-Nrxn1 SSDs

PSD-95

Merge

PSD-95 Normalized Density
 ≤50 100 ≥150

100nm

6g En Face

100nm

5. The authors found that more than 90% of excitatory synapses are Nr1 positive and there are averagely 2 Nr1 SSDs per synapse. These numbers are quite different from previous study using the same KI mice (Trotter 2019) showing that only 40% synapses have Nr1 and mose Nr1+ synapses have only one Nr1 nanocluter. How the authors intepret these differences? We believe that the differences in our observations are due to differences in the sensitivity of STORM methods for which we used direct STORM and Trotter et al. 2019 utilized dye pair STORM. The differences in the sensitivity of our two methods are most readily seen in their average number of GluA1 localizations for their control conditions in which they report a mean -4 localizations for synapses without HA-Nrxn1 and an average of -20 GluA1 localizations for synapses with HA-Nrxn1 (Trotter et al. 2019 6H-I). The total number of localizations within the PSD compartment characterized in Trotter et al., 2019 are much lower than the previously reported estimated values from multiple imaging modalities (ultrastructural and superresolution), which indicated that a single AMPAR SSD contains -20-25 molecules and the PSD compartment volume contains an average of -100 AMPARs at a typical synapse (Compans et al., 2016). Thus, the total number of GluA1 localizations detected in Trotter et al., 2019 in an entire PSD compartment volume is equal to the average number of GluA1 localizations in a single SSD, suggesting that the data may be under-sampled. Thus, the reported nanoscopic properties of HA-Nrxn1 may be a result of under-sampling. Using dSTORM methodology we observed a median 119 localizations per synapse, which is consistent with previous reports and supports the notion that we are sampling synapses adequately.

In addition to differences in STORM (dye-pair vs direct), neuron culture conditions differed between our study and Trotter et al., 2019. To test if differences in culture media composition explained the nanoscale differences, we assessed pan-Nrxn nanoscale properties in neurons grown in Neurobasal A media supplemented with B27 ("Serum Free;" Trotter et al., 2019) and Modified Eagles' Media supplemented with FBS and B27 (Serum Media) used in this study. We used the identical pan-Nrxn antibody from Trotter et al., 2019 to label all Nrxns. We found that media composition did not explain the differences Nrnx localizations. Most synapses (-90%), independent of culture conditions, contained one or more SSDs. As discussed in the manuscript, this finding that most synapses contain Nrnx SSDs is consistent with the effect sizes in multiple synapses following Nrnx manipulations (Missler et al., 2003; Aoto et al., 2015; Boxer et al., 2021; Hauser et al., 2022; Keum et al., 2018; Trotter et al., 2023; Chen et al., 2017; Anderson et al., 2015)

Effects of culture media on pan-Nrxn nanoscale organization

6. How is the enrichment index calculated?

We have included the following description of the enrichment index:

Nanocolumn alignment of two proteins would predict that the centroid of an SSD on one side of the synapse should oppose a region of higher normalized protein density on the other. We first quantitatively determined the number of GluA1 molecules within binned distances from the centroid of RIM1. We also randomized the GluA1 localizations and determined the number of randomized GluA1 molecules opposite RIM1 at the same binned distances. Dividing the experimentally determined density distribution by the randomized dataset resulted in the normalized GluA1 density from the centroid of a RIM1 SSD, thus, an enrichment value of 1 represents a randomized distribution of GluA1 molecules¹⁹ (Figure 1j). Further, we averaged the normalized GluA1 density within a radius of 60nm from the centroid of RIM1 to determine the enrichment index¹⁹ (Figure 1j).

Minors:

1. In abstract line 31, Nr3 could be the first presynaptic adhesion molecule identified to be required for nanocolumns but certainly not the first presynaptic molecule.

Corrected.

2. The order of citations were messed up. Some references, such as ref17, were never cited in the text.

Citations have been corrected.

3. Line 191, should be "the mean radial distance of V5-Nrxn3..." In the same sentence, the numbers represented in Fig 4C is missing.

Corrected.

4. I'm troubled by the overused "enrichment". For example, in line 250-251, there are two enrichments, one for a protein and the other for SSDs, both different from the context of "enrichment analysis/index".

We agree and use density as a more accurate description instead of enrichment. Where we use enrichment to primarily refer to measurements derived from the enrichment index.

Reviewer #2 (Remarks to the Author):

While we know much about the molecular machinery within each compartment, we are only beginning to understand how these compartments are structurally in register and functionally integrated with one another. It is likely that this architecture may allow for precise synaptic information exchange and may be modulated to contribute to the remarkable plasticity of brain function. Therefore this manuscript is timely and important. The main findings of this work, that NRXN1 and 3 are organized in specialized and distinct clusters, which allow them to retain their individual properties and function in parallel, are important to understand how synapses assemble and work.

Overall, this is an extremely data-rich manuscript with experiments carefully planned and executed. However, it is a bit hard to read, perhaps because most of the time the authors don't elaborate on what the measurements mean. For example in figure 1A-E, the authors describe variation in synaptic volume occupied by GluA1, number of GluA1 SSDs, GluA1 SSD volume, density, etc. Some parameters don't change while others do. But there is no explanation of what this all means, even if just structurally. Another example is at lines 152-157: Nr3 KO significantly alters the PSD-95 synaptic parameters. What conclusion can they draw from this that can drive the next set of experiments? More information to guide the reader on what each parameter/experiment is telling is crucial to follow the progression of experiments, otherwise the manuscript looks disjointed and a little descriptive.

We appreciate the positive feedback and hope that the reviewer appreciates that we have expanded the results and discussion to elaborate on our findings.

Introduction is well crafted and contains all the necessary bits to follow the experiments laid in

the result section.

Thank you.

Line 31 in the abstract (and line 70 and 125): what do the authors mean when they write that NRXN-1 is “the first presynaptic molecule required for excitatory synapse nano-organization”? That NRXN-1 is the first to be expressed, or that is “the first being identified” for the nano-organization? Please clarify.

We appreciate the comment and apologize for the confusion. We have altered these statements for clarity to:

Using Double Helix 3D dSTORM and new neurexin mouse models, we identify neurexin-3 as the first presynaptic adhesion molecule responsible for excitatory synapse nano-organization in hippocampus.

I presume NXN3 KO deletes both alpha and beta NRXN, but this is not stated, please specify.

Yes, we now specify that both isoforms (a□b) are deleted in the conditional knockout.

Perhaps the modelling of synaptic transmission (Figure 1) could be strengthened by actual ephys experiments using at least some of the different conditions utilized so that all of these SSDs findings can have a physiological interpretation.

We agree that changing different parameters of nano-organization and observing the electrophysiological phenotypes would be of great interest, however, methods to change single parameters of nano-organization have remained elusive. For example, in Ramsey et al. 2021, they cleaved an overexpressed LRRTM2 and it changed both AMPAR density in SSDs and reduced the number of GluA1 SSDs and in Han et al. 2022 Nlgn3 KO decreased the enrichment of AMPARs in SSDs but also increased the size of AMPAR SSDs. Interrogating the physiologic relevance of each of the properties of GluA1 SSDs represents an important line of inquiry but is currently not addressable due to the lack of available tools.

Figure 1L: Ablation of Nrnx3 reduced the density of GluA1 protein opposite RIM1 SSDs by more than 33%. Figure L shows a reduction from ~1.8 down to ~1.6, which is only ~12%. Or does this refer to figure 1D? Please explain the discrepancy

We apologize for the confusion. Figure 1l in the original submission did not refer to the enrichment index discussed in the Results Section but instead referred to the cross-correlation index. We have now included the calculated enrichment index in Figure 1i (control: ~2.3 and Nrnx3 KO: ~1.8). We understand that the readout of this particular analysis is not intuitive. For this analysis 1 is the average density in the synaptic compartment which means that a value of 1.8 represents 80% enrichment and 1.6 represents 60% enrichment for a mean difference of 25%. The 33% value is when comparing the average across the translation radii between 20-60nm. The enrichment index is determined as the normalized GluA1 density within a radius of 60nm from the centroid of RIM1 to determine the enrichment index.

Figure 2 e-g: what’s the difference between these graphs and the one shown in Figure 1 f-i? The analysis of RIM1 nanoscopic properties in Figures 1 and 2 were performed for completely separate sets of experiments separated by 1-2 months and highlights the reproducibility of our STORM imaging approaches. Further, we needed to co-label for PSD-95 and RIM1 to assess transsynaptic nanocolumn alignment.

I have no comments on the generation and testing of the V5 tagged NRXN mouse line, other than praising the authors for the effort in generating a new mouse line that will help understanding NRXN expression, function and physiology

We appreciate the positive comment.

Please explain why in figure 4, V5-Nrxn3 was first co-stained with Homer1 instead of PSD95, as they both identify excitatory synapses.

We used Homer1 first because it is an excitatory synapse marker from which a Homer1 has a relatively

homogenous distribution, in contrast to PSD-95, and therefore the Homer1 centroid is a reliable marker for an “origin” at excitatory synapses. For this reason, a variety of other synaptic proteins have been measured using super resolution relative to Homer1 (for example Dani et al. 2010).

While Homer1 is good for the purpose of providing an origin point at excitatory synapses, it is not a member of the synaptic nanocolumn (Tang et al. 2016). To both identify excitatory synapses and determine if V5-Nrxn3 localized to excitatory synaptic nanocolumns we then looked at the distribution of V5-Nrxn3 relative to PSD-95.

Figures need to be numbered on the file, otherwise it's too difficult to follow which experiments is referred to in the text.

Line 278: please explain what do you mean by “non-redundant transsynaptic signaling.”

For clarity, we have changed the sentence to:

Our data raise the intriguing possibility that the spatial nanoscale segregation of Nrxn1 and Nrxn3 may be positioned to engage in parallel transsynaptic signaling pathways through GluD1 and LRRTM2, respectively.

Finally, an overall schematic model of the findings would be helpful to wrap it all up.

We have added a cartoon schematic of our findings – Figure 9.

Reviewer #3 (Remarks to the Author):

Neurexins (Nrxns) are presynaptic adhesion molecules that bind to multiple postsynaptic partners to control the specification of various types of central synapses. Nrxns have a large number of isoforms that distinctly interact with different postsynaptic binding partners. Importantly, the repertoire of presynaptic Nrxn isoforms and their postsynaptic binding partners at individual synapses differ synapse by synapse, depending on cell types, neuronal circuits, and brain regions at tissue levels.

However, how this diverse repertoire specifies the function of individual synapses largely remains unknown. The authors address this important question by nano-scale molecular imaging of hippocampal primary cultures with super-resolution microscopy. The authors demonstrated that Nrxn3, but not Nrxn1, was required for trans-synaptic organizations of RIM1 and AMPAR nanoclusters at excitatory synapses, which can play an important role in glutamatergic transmission. By generating a new mouse line expressing both HA-tagged Nrxn1 and V5-tagged Nrxn3, the authors also revealed that presynaptic nanoclusters of Nrxn1 and Nrxn3 were segregated from each other, and distinctly arranged close proximity to postsynaptic nanoclusters of GluD1 and LRRTM2, respectively. Their experimental technique is state of art, and nicely visualizes subsynaptic nano-scale clusters of synaptic proteins on the presynaptic and postsynaptic membrane. However, this study seems to be descriptive because it lacks a mechanistic explanation to the segregated distribution of Nrxn1 and Nrxn3 nanoclusters on the presynaptic membrane. For instance, does this segregation depends on the difference between Nrxn1 and Nrxn3 isoforms themselves or dominant recruitments of Nrxn1+SS4(interacts with Cbln-GluD1) and Nrxn3-SS4(interacts with LRRTM2) into Nrxn1 and Nrxn3 nanoclusters, respectively? It also remains unclear whether Nrxn1 and Nrxn3 nanoclusters actually interact with GluD1 and LRRTM2 nanoclusters in a segregated fashion. In addition, the authors should address the following specific points.

We appreciate the positive feedback on our manuscript that our findings address important findings in the field. Our work represents an important first step toward understanding the mechanisms that permit individual neurexins to control distinct aspects of neurotransmission. We agree that it will be important to identify the mechanism(s) that spatially segregate Nrxn1 and Nrxn3 SSDs, however, it is beyond the scope of the current manuscript. As discussed in the Discussion section, we do not believe that the spatial segregation is dependent on the inherent SS4 identity of Nrxns. Both Nrxn1 and Nrxn3 are primarily expressed in hippocampus and cortex without the SS4 exon (SS4–) (Aoto et al., 2013;

Schreiner et al., 2015; Lukacsovich et al., 2019; Nguyen et al., 2016, but see: Trotter et al., 2023). If Nrnxn segregate based on SS4 expression, we would predict that in brain regions like hippocampus where SS4 is preferentially excluded, that Nrnxn SSDs would exhibit high degrees of overlap. Although the ligand binding affinities are not commonly performed on multiple Nrnxns, the ligands that have been tested for binding with Nrnxn1, Nrnxn2 and Nrnxn3 (e.g. Cbln and LRRTM2) exhibit higher binding affinities for Nrnxn1 relative to Nrnxn3. However, we acknowledge that binding assays performed *in vitro* or in heterologous cell systems, while informative, may not truly reflect binding preferences at the synaptic cleft and cannot completely exclude this possibility. We hypothesize that intracellular binding partners that differentially interact with Nrnxn1 and Nrnxn3 may contribute to the nanoscopic spatial segregation observed here.

We agree that the nanoscale radial distributions of Nrnxn3 and LRRTM2, and, to a lesser extent Nrnxn1 and GluD1, suggest that they may be transsynaptic oriented to potentially interact. However, testing for transsynaptic interactions of endogenous proteins at synapses has proven elusive. Approaches using reconstituted GFP (e.g. mGRASP) or biotin ligation (e.g. BLINC, TurboID), while powerful tools, require either overexpression of exogenous ligand/receptor and/or insertion of foreign DNA sequences into proteins of interest. The potential caveats and interpretation of the results due to the overexpression of exogenous protein and/or the potential for the disruption in protein folding/trafficking limit our enthusiasm for these approaches.

#1 The authors' conclusion is limited in scope because this study tested hippocampal culture synapses only. To make the current conclusion more generalized, the authors should test different types of synapses with distinct repertoires of Nrnxns in other neuronal cultures or brain tissues. In this respect, the statement "Nrnxn3 is the first functionally-relevant presynaptic adhesion molecule responsible for organizing the nanoscale architecture of excitatory synapses" sounds too strong to me.

To address this valid statement, we have performed key experiments in cortex to assess if the findings reported in hippocampus are generalizable to other brain regions. The impact of Nrnxn3 KO on excitatory transmission on cortical neurons has not been previously assessed. However, we did not observe a nanoscale GluA1 phenotype in cortical cultures (Figure 2h-n and S2m-o). As discussed, and elaborated on in the manuscript, this could be due to a number of reasons including:

1. Nrnxn3 does not control AMPAR-mediated synaptic transmission in cortical neurons.
2. Nrnxn3 controls different synaptic properties in hippocampus and cortex. This is not unreasonable as Nrnxn3 is pleiotropic and its function seems to vary depending on the brain region and cell-type studied.
3. Unlike hippocampus, Nrnxn3a and Nrnxn3b are not uniformly expressed at high levels in the different cortical layers. Moreover, Nrnxn3 is highly expressed in GABAergic neurons in cortex. The heterogeneous expression of Nrnxn3 as well as the cellular heterogeneity of cortical cultures might mask an AMPAR phenotype in the Nrnxn3 KO.

Importantly the nanoscale segregation of Nrnxn1 and Nrnxn3 SSDs persists in cortical neurons, suggesting that this segregation may be a more general nanoscale property of these two adhesion molecules (Figure s8n-r).

#2 GluD1 may have not only the cell-surface pool but also the intracellular reserve one, as in the case of GluAs. However, the current data cannot distinguish the former from the latter at individual synapses, making it difficult to judge signals presented in this study as the cell-surface GluD1 nanoclusters. The authors should test the cell-surface GluD1 by surface labeling with antibodies against its extracellular domain or demonstrate the current signals are selectively associated with the cell-surface pool, but not the reserve pool.

We agree that the intracellular antibodies for GluD1 cannot distinguish the surface vs intracellular pools of the receptor and it is impossible to surface nanoclusters. Unfortunately, we only found one antibody that claims to detect extracellular epitopes of GluD1 (Alomone AGC-038). We tested the antibody using the recommended dilution range and found that the antibody is not specific. We compared a region in which there were dendrites and glia (left panels) to a region lacking any neurons or glia (e.g. glass

coverslip only; second from the left) and observed no difference in signal. This non-specific signal was not a result of non-specific secondary antibodies as the secondary antibody alone produced minimal signal.

Only available surface epitope GluD1 exhibits no selectivity for cells

We include the following caveat in the results section following our GluD1 analysis:

However, the use of an intracellular antibody does not distinguish between single molecule localizations representing surface or intracellular GluD1, and likely underestimates the actual radial distances of the extracellular sequences from Homer1 that mediate the tripartite GluD1-Cbln-Nrxn complex.

#3 The validation of anti-LRRTM2 antibody is not convincing. The current data cannot rule out a possibility that the remaining labeled puncta in LRRTM2-knockdown neurons are non-specific signals. The authors should validate this antibody with more appropriate controls such as LRRTM2-KO mice.

Currently, the validated LRRTM2 cKO originally characterized in Bhourri et al., 2018 is not commercially available through Jackson Labs.

We remade the lentivirus encoding the previously published LRRTM2 shRNA sequences and observed a significant reduction in the levels of mRNA and surface LRRTM2 protein levels (Figure s7a-c).

LRRTM2 shRNA reduced mRNA levels by 91% and surface protein by ~85%. shRNA suppresses but does not eliminate expression, our data suggest that non-specific binding of the LRRTM2 antibody

#4 Although this study is highly dependent on immunocytochemistry, the current manuscript also lacks the information about the specificity of immunocytochemical signals. The authors can present more information about the signal specificity under the authors' experimental conditions.

We now include details for the validated commercially available antibodies used here. In addition to validating the LRRTM2 antibody from BLOSS, we also validated the BLOSS anti-GluD1 antibody. We first attempted to measure protein levels after knockdown using a previously published shRNA (Fossatti et al., 2019), however, in our hands, shRNA knockdown did not consistently deplete endogenous GluD1 by Western blot.

Published GluD1 shRNA shows minimal GluD1 protein knockdown

Instead, we repeated 3D dSTORM with a KO validated antibody GluD1⁸⁹⁵⁻⁹³² (Konno et al., 2014; Frontier Institute: GluD1C-GP-Af860). Similar to the GluD1 antibody used in the initial submission anti-GluD1⁵⁰¹⁻⁶⁰⁰ (BLOSS), the KO validated antibody detects intracellular sequences of GluD1, however, the KO antibody is not amenable to direct conjugation with a fluorophore. We compared our direct GluD1

3D STORM against anti-GluD1⁸⁹⁵⁻⁹³². Figure S7f-m details how the nanoscale properties (numbers of localizations and radial distributions relative to Homer1) for the KO validated antibody are virtually identical to the BIOSS GluD1 antibody. We now include both antibodies for our nanoscale characterization of GluD1. Since the radial distributions from the centroid of Homer1 are nearly identical for the single molecule localizations and SSDs of GluD1, we kept the nearest neighbor analysis from the initial submission, which indicated that the nearest neighbor distances for GluD1-Nrxn1 are considerably shorter when compared to the distances for GluD1-Nrxn3.

References

1. Anderson, G.R., Aoto, J., Tabuchi, K., Foldy, C., Covy, J., Yee, A.X., Wu, D., Lee, S.J., Chen, L., Malenka, R.C., and Sudhof, T.C. (2015). beta-Neurexins Control Neural Circuits by Regulating Synaptic Endocannabinoid Signaling. *Cell* *162*, 593-606. 10.1016/j.cell.2015.06.056.
2. Bhourri, M., Morishita, W., Temkin, P., Goswami, D., Kawabe, H., Brose, N., Sudhof, T.C., Craig, A.M., Siddiqui, T.J., and Malenka, R. (2018). Deletion of LRRTM1 and LRRTM2 in adult mice impairs basal AMPA receptor transmission and LTP in hippocampal CA1 pyramidal neurons. *Proc Natl Acad Sci U S A* *115*, E5382-E5389. 10.1073/pnas.1803280115.
3. Boxer, E.E., Seng, C., Lukacsovich, D., Kim, J., Schwartz, S., Kennedy, M.J., Foldy, C., and Aoto, J. (2021). Neurexin-3 defines synapse- and sex-dependent diversity of GABAergic inhibition in ventral subiculum. *Cell Rep* *37*, 110098. 10.1016/j.celrep.2021.110098.
4. Chamma, I., Letellier, M., Butler, C., Tessier, B., Lim, K.H., Gauthereau, I., Choquet, D., Sibarita, J.B., Park, S., Sainlos, M., and Thoumine, O. (2016). Mapping the dynamics and nanoscale organization of synaptic adhesion proteins using monomeric streptavidin. *Nat Commun* *7*, 10773. 10.1038/ncomms10773.
5. Chen, L.Y., Jiang, M., Zhang, B., Gokce, O., and Sudhof, T.C. (2017). Conditional Deletion of All Neurexins Defines Diversity of Essential Synaptic Organizer Functions for Neurexins. *Neuron* *94*, 611-625 e614. 10.1016/j.neuron.2017.04.011.
6. Cohen, R.N., van der Aa, M.A., Macaraeg, N., Lee, A.P., and Szoka, F.C., Jr. (2009). Quantification of plasmid DNA copies in the nucleus after lipoplex and polyplex transfection. *J Control Release* *135*, 166-174. 10.1016/j.jconrel.2008.12.016.
7. Compans, B., Choquet, D., and Hosy, E. (2016). Review on the role of AMPA receptor nano-organization and dynamic in the properties of synaptic transmission. *Neurophotonics* *3*, 041811. 10.1117/1.NPh.3.4.041811.
8. Diaz-Alonso, J., Sun, Y.J., Granger, A.J., Levy, J.M., Blankenship, S.M., and Nicoll, R.A. (2017). Subunit-specific role for the amino-terminal domain of AMPA receptors in synaptic targeting. *Proc Natl Acad Sci U S A* *114*, 7136-7141. 10.1073/pnas.1707472114.
9. Guzikowski, N.J., and Kavalali, E.T. (2021). Nano-Organization at the Synapse: Segregation of Distinct Forms of Neurotransmission. *Front Synaptic Neurosci* *13*, 796498. 10.3389/fnsyn.2021.796498.
10. Hauser, D., Behr, K., Konno, K., Schreiner, D., Schmidt, A., Watanabe, M., Bischofberger, J., and Scheiffele, P. (2022). Targeted proteoform mapping uncovers specific Neurexin-3 variants required for dendritic inhibition. *Neuron* *110*, 2094-2109 e2010. 10.1016/j.neuron.2022.04.017.
11. Konno, K., Matsuda, K., Nakamoto, C., Uchigashima, M., Miyazaki, T., Yamasaki, M., Sakimura, K., Yuzaki, M., and Watanabe, M. (2014). Enriched expression of GluD1 in higher brain regions and its involvement in parallel fiber-interneuron synapse formation in the cerebellum. *J Neurosci* *34*, 7412-7424. 10.1523/JNEUROSCI.0628-14.2014.
12. Lukacsovich, D., Winterer, J., Que, L., Luo, W., Lukacsovich, T., and Foldy, C. (2019). Single-Cell RNA-Seq Reveals Developmental Origins and Ontogenetic Stability of Neurexin Alternative Splicing Profiles. *Cell Rep* *27*, 3752-3759 e3754. 10.1016/j.celrep.2019.05.090.
13. Missler, M., Zhang, W., Rohlmann, A., Kattenstroth, G., Hammer, R.E., Gottmann, K., and Sudhof, T.C. (2003). Alpha-neurexins couple Ca²⁺ channels to synaptic vesicle exocytosis. *Nature* *423*, 939-948. 10.1038/nature01755.
14. Nair, D., Hosy, E., Petersen, J.D., Constals, A., Giannone, G., Choquet, D., and Sibarita, J.B. (2013). Super-resolution imaging reveals that AMPA receptors inside synapses are dynamically organized in nanodomains regulated by PSD95. *J Neurosci* *33*, 13204-13224. 10.1523/JNEUROSCI.2381-12.2013.
15. Nguyen, T.M., Schreiner, D., Xiao, L., Traunmuller, L., Bornmann, C., and Scheiffele, P. (2016). An alternative splicing switch shapes neurexin repertoires in principal neurons versus interneurons in the mouse hippocampus. *Elife* *5*. 10.7554/eLife.22757.
16. Ramsey, A.M., Tang, A.H., LeGates, T.A., Gou, X.Z., Carbone, B.E., Thompson, S.M., Biederer, T., and Blanpied, T.A. (2021). Subsynaptic positioning of AMPARs by LRRTM2 controls synaptic strength. *Sci Adv* *7*. 10.1126/sciadv.abf3126.

17. Schreiner, D., Simicevic, J., Ahrne, E., Schmidt, A., and Scheiffele, P. (2015). Quantitative isoform-profiling of highly diversified recognition molecules. *Elife* 4, e07794. 10.7554/eLife.07794.
18. Tang, A.H., Chen, H., Li, T.P., Metzbower, S.R., MacGillavry, H.D., and Blanpied, T.A. (2016). A trans-synaptic nanocolumn aligns neurotransmitter release to receptors. *Nature* 536, 210-214. 10.1038/nature19058.
19. Trotter, J.H., Wang, C.Y., Zhou, P., Nakahara, G., and Sudhof, T.C. (2023). A combinatorial code of neurexin-3 alternative splicing controls inhibitory synapses via a trans-synaptic dystroglycan signaling loop. *Nat Commun* 14, 1771. 10.1038/s41467-023-36872-8.

REVIEWERS' COMMENTS

Reviewer #1 (Remarks to the Author):

All my major concerns have been addressed in the revised ms. I appreciate the amount of data they authors have presented in this study.

One minor: In part of figure 2 related to PSD-95, the authors should report information on the relative PSD-95 density in SSD as they did for GluA1.

Reviewer #2 (Remarks to the Author):

I'd like to thank the authors for satisfying all my concerns. I have not other question.

Reviewer #3 (Remarks to the Author):

I appreciate the authors'effort to improve the manuscript in generality and reliability. Although the authors'responses largely convinced me, I still feel the manuscript remains a little bit too descriptive with regard to parallel transsynaptic signaling mediated by Nrxn1/Nrxn3 SSDs because of the lack of direct evidence for functional interactions between Nrxn1/Nrxn3 SSDs and their postsynaptic binding partners at synapses. It could be worth showing the presence or absence of LRRTM2 or GluD1 nanoclusters in Nrxn1-KO or Nrxn3-KO, if the authors have such results.

Reviewer #1 (Remarks to the Author):

All my major concerns have been addressed in the revised ms. I appreciate the amount of data they authors have presented in this study.

Thank you for the positive feedback.

One minor: In part of figure 2 related to PSD-95, the authors should report information on the relative PSD-95 density in SSD as they did for GluA1.

We have now included our measurements of PSD95 molecules within SSDs as Figure S2a.

There is no change in PSD-95 density in SSDs after Nrnx3 KO:

Reviewer #2 (Remarks to the Author):

I'd like to thank the authors for satisfying all my concerns. I have not other question. Thank you very much.

Reviewer #3 (Remarks to the Author):

I appreciate the authors' effort to improve the manuscript in generality and reliability. Although the authors' responses largely convinced me, I still feel the manuscript remains a little bit too descriptive with regard to parallel transsynaptic signaling mediated by Nrnx1/Nrnx3 SSDs because of the lack of direct evidence for functional interactions between Nrnx1/Nrnx3 SSDs and their postsynaptic binding partners at synapses. It could be worth showing the presence or absence of LRRTM2 or GluD1 nanoclusters in Nrnx1-KO or Nrnx3-KO, if the authors have such results.

We appreciate the reviewer's statements that our responses have largely convinced the reviewer and that we have improved the manuscript.

We appreciate the suggested experiment proposed by the reviewer, however, we do not have results readily available. Assaying the nanoscale properties of GluD1 and LRRTM2 in Nrnx1 and Nrnx3 KO neurons serves as a proxy for possible transsynaptic interactions, however, the design of the experiment makes the critical assumption that Nrnx1 and Nrnx3 anchor GluD1 and LRRTM2, respectively, to their subsynaptic regions. How GluD1 and LRRTMs form SSDs and exhibit their subsynaptic localization is unknown and represent an important topic for future studies. We have previously shown that the forced genetic inclusion of splice-site #4 (SS4+),

which cannot bind LRRTMs, reduced the surface abundance of LRRTM2 (Aoto et al., 2013). It is unclear whether the loss of surface LRRTM2 was due to the direct loss of SS4– dependent transsynaptic interactions with LRRTM2 or due to an indirect, and yet uncharacterized, mechanism driven by SS4+. Further, synaptic compartment volumes and SSDs properties are not inextricably linked (e.g. RIM1 in *Nrxn3* KO neurons and GluA1 properties following the manipulation of LRRTM2 (Ramsey et al., 2021)). SSD properties of LRRTM2 were not assessed in *Nrxn3* SS4 KI neurons. The nature of the *Nrxn3* SS4 KI mutant mouse (Aoto et al., 2013) is fundamentally distinct from the *Nrxn3* KO mouse (Aoto et al., 2015) in that *Nrxn3* protein level is unaltered in the SS4 KI but is eliminated in the KO. While direct binding with *Nrxns* may stabilize GluD1 and LRRTM2 SSD maintenance and/or subsynaptic localization, these nanoscopic parameters may be independent of transsynaptic interactions and/or may be indirectly stabilized via *Nrxn1/3*. Thus, no change in SSD properties of GluD1 and/or LRRTM2 in *Nrxn* KO neurons does not exclude the possibility that *Nrxn*-ligand binding is disrupted. Conversely, an observed change in ligand SSD properties after *Nrxn1* or *Nrxn3* KO does not necessarily demonstrate direct binding.